# Estimation of the Antarctic surface mass balance using the regional climate model MAR (1979-2015) and identification of dominant processes

Cécile Agosta[1,2,3], Charles Amory[1], Christoph Kittel[1], Anais Orsi[2], Vincent Favier[3], Hubert Gallée[3], Michiel R. van den Broeke[4], Jan T. M. Lenaerts[4,5], Jan Melchior van Wessem[4], Willem Jan van de Berg[4], and Xavier Fettweis[1]

[1]F.R.S.-FNRS, Laboratory of Climatology, Department of Geography, University of Liège, B-4000 Liège, Belgium
[2]Laboratoire des Sciences du Climat et de l'Environnement (IPSL/CEA-CNRS-UVSQ UMR 8212), CEA Saclay, F-91190 Gif-sur-Yvette, France
[3]Université Grenoble Alpes, CNRS, Institut des Géosciences de l'Environnement, F-38000, Grenoble,France
[4]Institute for Marine and Atmospheric Research Utrecht, Utrecht University, Utrecht, the Netherlands
[5]Department of Atmospheric and Oceanic Sciences, University of Colorado Boulder, Boulder CO, United States of America

**Correspondence:** Cécile Agosta (cecile.agosta@gmail.com)

**Abstract.** The Antarctic ice sheet mass balance is a major component of the sea level budget and results from the difference of two fluxes of a similar magnitude: ice flow discharging in the ocean and net snow accumulation on the ice sheet surface, i.e. the surface mass balance (SMB). Separately modelling ice dynamics and surface mass balance is the only way to project future trends. In addition, mass balance studies frequently use regional climate models (RCMs) outputs as an alternative to observed fields because SMB observations are particularly scarce on the ice sheet. Here we evaluate new simulations of the polar RCM MAR forced by three reanalyses, ERA-Interim, JRA-55 and MERRA2, for the period 1979-2015, and we compare MAR results to the last outputs of the RCM RACMO2 forced by ERA-Interim. We show that MAR and RACMO2 perform similarly well in simulating coast to plateau SMB gradients, and we find no significant differences in their simulated SMB when integrated over the ice sheet or its major basins. More importantly, we outline and quantify missing or underestimated processes in both RCMs. Along stake transects, we show that both models accumulate too much snow on crests, and not enough snow in valleys, as a result of drifting snow transport fluxes not included in MAR and probably underestimated in RACMO2 by a factor of three. Our results tend to confirm that drifting snow transport and sublimation fluxes are much larger than previous model-based estimates and need to be better resolved and constrained in climate models. Sublimation of precipitating particles in low-level atmospheric layers is responsible for the significantly lower snowfall rates in MAR than in RACMO2 in katabatic channels at the ice sheet margins. Atmospheric sublimation in MAR represents 363 Gt yr$^{-1}$ over the grounded ice sheet for the year 2015, which is 16 % of the simulated snowfall loaded at the ground. This estimate is consistent with a recent study based on precipitation radar observations, and is more than twice as much as simulated in RACMO2, because of different time residence of precipitating particles in the atmosphere. The remaining spatial differences in snowfall between MAR and RACMO2 are attributed to differences in advection of precipitation, snowfall particles being likely advected too far inland in MAR.

# 1   Introduction

Mass loss from the Antarctic ice sheet (AIS) and therewith its contribution to the sea level budget results from the difference of two fluxes of a similar magnitude: ice flow discharging in the ocean (D) and net snow accumulation on the ice sheet surface, i.e. the surface mass balance (SMB). The total ice sheet mass balance (SMB minus D) can be assessed using satellite altimetry, gravimetry or the input–output method (Shepherd et al., 2018), which all request surface mass balance estimates. The input-output method, which consists in separately modelling ice dynamics and surface mass balance, is also the only way to project future trends.

Surface mass balance as used in this study is the sum of mass gains (mainly snowfall accumulation and some riming), mass losses (mainly surface and drifting snow sublimation, some liquid water runoff) and drifting snow transport (defined as the horizontal advection of the drifting snow) which can lead to either mass gain or mass loss. Snowfall rates are one order of magnitude larger than all of the other SMB fluxes at the continental scale (Lenaerts et al., 2012b), with the largest amounts found along the ice sheet margins due to cyclonic activity in the Southern Ocean and to the orographic lifting of relatively warm and moist air masses (van Wessem et al., 2014; Favier et al., 2017). Accumulation patterns are highly variable at the kilometre scale and from year to year (e.g., Agosta et al., 2012). Consequently, proper observations of SMB require a high spatial coverage (e.g. stake-lines, accumulation radars plus ice-cores for layer dating and snow density) and a temporal sampling spanning several years (Eisen et al., 2008). Even if efforts have been made to fulfil those requirements, ground-based observations are scarce and carry with them high logistical costs in this cold, windy and remote environment. Interpolation techniques used to interpolate the scarce SMB observations (Vaughan et al., 1999; Arthern et al., 2006) encounter major caveats (Magand et al., 2008; Genthon et al., 2009; Picard et al., 2009).

This is why many AIS mass balance studies use output of regional climate models (RCMs) to estimate ice sheet SMB for the recent decades (e.g., Rignot et al., 2011; Gardner et al., 2018; Shepherd et al., 2018). In order to obtain a good agreement with observations, atmospheric models require accurate large-scale circulation patterns together with a proper representation of snow surface processes, clouds, turbulent fluxes, and a relatively high horizontal resolution to properly resolve the complex ice sheet topography at the margins.

Here, we present new simulations of the regional climate model MAR, applied for the first time over the whole AIS, but already widely used for polar studies, e.g. in Greenland (Fettweis et al., 2013, 2017), Svalbard (Lang et al., 2015), Adélie Land (Antarctic coastal area, Gallée et al., 2013; Amory et al., 2015) and Dome C (Antarctic plateau, Gallée et al., 2015). We compare MAR-simulated SMB with the state-of-the-art regional climate model RACMO2 (van Wessem et al., 2018). We use available SMB observational datasets to show that MAR and RACMO2 perform similarly well in simulating the SMB spatial gradients. In addition, we identify significant processes that still need to be included or improved in both RCMs.

In Section 2, we describe MAR and its specific set-up for Antarctica, together with RACMO2, the forcing fields, observational datasets and methods designed for model evaluation. In Section 3, we show that both RCMs share common biases against observed SMB, resulting from drifting snow transport fluxes. Secondly, we analyse SMB differences between models and show that many of the discrepancies can be attributed to low-level sublimation of precipitation in katabatic channels and

to the difference in precipitation advection inland. Finally, in Section 4, we summarise our main findings and discuss further efforts to be achieved for a better assessment of the AIS surface mass balance.

## 2 Data and methods

### 2.1 Regional modelling

#### 2.1.1 Regional atmospheric models

For the first time, the polar-oriented regional atmospheric model MAR is applied for decades-long simulations over the whole Antarctic ice sheet. MAR atmospheric dynamics are based on the hydrostatic approximation of the primitive equations, fully described in Gallée and Schayes (1994). Prognostic equations are used to depict five water species: specific humidity, cloud droplets and ice crystals, raindrops and snow particles (Gallée, 1995). Sublimation of airborne snow particles is a direct contribution to the heat and moisture budget of the atmospheric layer in which these particles are simulated. The radiative transfer through the atmosphere is parametrised as in Morcrette (2002), with snow particles affecting the atmospheric optical depth (Gallée and Gorodetskaya, 2010). The atmospheric component is coupled to the surface scheme SISVAT (soil ice snow vegetation atmosphere transfer, De Ridder and Gallée, 1998) dealing with the energy and mass exchanges between surface, snow and atmosphere. The snow–ice part of SISVAT is based on the snow model CROCUS (Brun et al., 1992). It is a one-dimensional multilayered energy balance model which simulates meltwater refreezing, snow metamorphism and snow surface albedo depending on snow properties. We used MAR version 3.6.4, simply called MAR here-after. In this version the physical settings are the same as in MAR version 3.5.2 used for Greenland (Fettweis et al., 2017), except for the adaptations detailed below.

*Grid:* Projection is the standard Antarctic polar stereographic (EPSG:3031). The horizontal resolution is 35 km, an intermediate resolution that results from a computation time compromise in order to run the model with multiple reanalyses and global climate model forcings over the 20th and the 21st century. The vertical discretisation is composed of 23 hybrid levels from $\sim$2 m to $\sim$17000 m above the ground.

*Boundaries:* The topography is derived from the Bedmap2 surface elevation dataset (Fretwell et al., 2013). Because the Antarctic domain is about 4 times larger than the Greenland domain, the circulation has to be more strongly constrained. This is why we use a boundary relaxation of temperature and wind in the upper atmosphere starting from 400 hPa ($\sim$6000 m above the ground) to 50 hPa (upper level), as in van de Berg and Medley (2016), whereas relaxation starts from 200 hPa in Fettweis et al. (2017).

*Parameterisations:*

   a) The surface snow density $\rho_s$ (kg m$^{-3}$) is computed as a function of 10 m wind speed $ws_{10}$ (m s$^{-1}$) and surface temperature $T_s$ (K):

$$\rho_s = 149.2 + 6.84\, ws_{10} + 0.48\, T_s, \tag{1}$$

with minimum-maximum values of 200–400 $\mathrm{kg\,m^{-3}}$. This parameterisation was defined so that the simulated density of the first 50 cm of snow fits observations collected over the Antarctic ice sheet (see Fig. S1, with snow density database detailed in Table S1).

b) The aerodynamic roughness length $z_0$ is computed as a function of the air temperature, as proposed in Amory et al. (2017). The parameterisation was tuned so that $z_0$ fit the observed seasonal variation between high (> 1 mm) summer and lower (0.1 mm) winter values in coastal Adélie Land, for air temperatures above -20 °C. For lower temperatures, $z_0$ is kept constant and set to 0.2 mm, in agreement with observed $z_0$ values on the Antarctic Plateau (e.g., Vignon et al., 2016);

c) As in Fettweis et al. (2017), the MAR drifting snow scheme is not activated, because this scheme was sensitive to parameter choices (Amory et al., 2015). An updated version of the drifting snow scheme is currently being developed and evaluated for application at the scale of the whole ice sheet.

We compare MAR results over the AIS to the latest outputs of the regional atmospheric model RACMO2 version 2.3p2 (van Wessem et al., 2018), called RACMO2 here-after, using a horizontal resolution of 27 km, a vertical resolution of 40 atmospheric levels, and a topography based on the digital elevation model from Bamber et al. (2009). This regional model is developed by the Royal Netherlands Meteorological Institute (KNMI), and has subsequently been adapted for modelling the Antarctic climate and its surface mass balance (van de Berg et al., 2006). It includes a drifting snow scheme (Lenaerts et al., 2012a), an albedo routine with prognostic snow grain size (Kuipers Munneke et al., 2011), and a multilayer snow model computing melt, percolation, refreezing and runoff (Ettema et al., 2010).

MAR and RACMO2 models were developed independently. We will not detail here the many physical parameterisation differences between both RCMs, but we will later highlight some of them we show having a significant impact on the modelled SMB.

### 2.1.2 Forcing reanalyses

Regional atmospheric models are forced by atmospheric fields at their lateral boundaries (pressure, wind, temperature, humidity), at the top of the troposphere (temperature, wind), as well as by sea surface conditions (sea ice concentration, sea surface temperature) every six hours. Consequently, regional atmospheric models add details and physics to the forcing model in the mid and lower troposphere and at the land or iced surface, whereas large-scale circulation patterns are driven by the forcing fields. We forced MAR with three reanalyses over Antarctica in order to evaluate the uncertainty in the simulated surface climate arising from the uncertainty in the assimilation systems: the European Centre for Medium-Range Weather Forecasts "Interim" re-analysis (here-after ERA-Interim, resolution ∼0.75°, i.e. ∼50 km at 70 °S, Dee et al., 2011), the Modern-Era Retrospective analysis for Research and Applications Version 2 (here-after MERRA2, resolution ∼0.5°, Gelaro et al., 2017), and the Japanese 55-year Reanalysis from the Japan Meteorological Agency (here-after JRA-55, resolution ∼1.25°, Kobayashi et al., 2015).

The regional atmospheric model RACMO2 is forced by ERA-Interim. We focus our study to the period 1979-2015, as reanalyses are known to be unreliable before 1979, when satellite sounding data started to be assimilated (Bromwich et al., 2007).

## 2.2 Observations

### 2.2.1 SMB observations and sectors of strong SMB gradients

We use surface mass balance observations of the GLACIOCLIM-SAMBA dataset detailed in Favier et al. (2013) and updated by Wang et al. (2016). This dataset is an update of the one assembled by Vaughan et al. (1999) following the quality-control methodology defined by Magand et al. (2007). It includes 3043 reliable SMB values averaged over more than 3 years. We add accumulation estimates from Medley et al. (2014), retrieved over the Amundsen Sea coast (Marie Byrd Land) with an airborne-radar method combined with ice-core glaciochemical analysis.

The first order feature of the Antarctic SMB is a strong coastal-inland gradient, with mean values ranging from typically greater than $500 \, \text{kg} \, \text{m}^{-2} \, \text{yr}^{-1}$ at the ice sheet margins to about $30 \, \text{kg} \, \text{m}^{-2} \, \text{yr}^{-1}$ in the dry interior plateau (Fig. 1a, see also, e.g., Wang et al., 2016). As observations only cover 5 % of MAR grid cells over the ice sheet, we divide that sparse observation dataset into 10 sectors detailed in Table 1 and shown in Fig. 2. Six of them are stake transects with a stake every ~1.5 km, which have been proven very valuable for evaluating modelled SMB (Agosta et al., 2012; Favier et al., 2013; Wang et al., 2016). The four other sectors are composed of more scattered observations covering large elevation ranges (Victoria Land, Dronning Maud Land, and Ross Ice Shelf–Marie Byrd Land).

### 2.2.2 Model-observation comparison method

5    RACMO2 outputs are bi-linearly interpolated to the 35×35 km MAR grid. For each SMB observation, we consider the 4 surrounding MAR grid cells, from which we eliminate ocean grid cells. We also eliminate surrounding grid cells with an elevation difference with the observation greater than 200 m (missing elevation of observation is set to Bedmap2 elevation at 1 km resolution). Finally, we bi-linearly interpolate model values of the remaining grid cells at the observation location (see schematic in Fig. S2).

As we restrict our modelling study to the 1979-2015 period, we only consider observations beginning after 1950. For observations beginning after 1979, we time-average model outputs for the same period as the observation. We keep observations beginning before 1979 only if they cover more than eight years, and in this case we compare the observed value with the
5    modelled value time-averaged for 1979-2015.

In a last step, we average-out the kilometre-scale variability of the observed SMB (Agosta et al., 2012) by binning point values onto grid cells. For each grid cell containing multiple observations, we average all observations contained into the grid cell weighted by the time span of observations, and in the same way we weight-average the modelled values interpolated to observation locations. This way, we obtain consistent observed and modelled averaged values on grid cells.

**Table 1.** Sectors extracted from the GLACIOCLIM-SAMBA database.

| Sector name | Sector type | Nb. of obs. | Nb. of grid cells | Year range | Elevation range (m a.s.l.) | Ref. |
|---|---|---|---|---|---|---|
| Marie Byrd Land | Radar transects | 6615 | 57 | 1980–2009 | 973–1873 | [1] |
| Ross–Marie Byrd Land | Scattered | 72 | 51 | 1950–1991 | 37–1995 | [2,3,4] |
| Victoria Land | Scattered | 60 | 40 | 1951–2006 | 1804–3240 | [5,6,7] |
| Dumont-d'Urville–Dome C | Transect | 116 | 24 | 1955–2010 | 633–3240 | [5,8,9,10] |
| Law Dome–Wilkes Land | Transect | 382 | 32 | 1973–1986 | 801–2232 | [11] |
| Zhongshan–Dome A | Transect | 583 | 40 | 1994–2011 | 1031–4081 | [12,13] |
| Mawson–Lambert Glacier | Transect | 515 | 36 | 1990–1995 | 1883–2924 | [14] |
| Syowa–Dome F | Transect | 507 | 38 | 1955–2010 | 584–3803 | [15] |
| Princ. Elisabeth | Transect | 58 | 6 | 2009–2012 | 47–1071 | [16] |
| Dronning Maud Land | Scattered | 376 | 104 | 1955–2008 | 1753–3741 | [17,18,19,20] |

[1] Medley et al. (2014), [2] Clausen et al. (1979), [3] Venteris and Whillans (1998), [4] Vaughan et al. (1999), [5] Magand et al. (2007), [6] Frezzotti et al. (2004), [7] Frezzotti et al. (2007), [8] Pettré et al. (1986), [9] Agosta et al. (2012), [10] Verfaillie et al. (2012), [11] Goodwin (1988), [12] Ding et al. (2011), [13] Wang et al. (2016), [14] Higham and Craven (1997), [15] Wang et al. (2015), [16] GLACIOCLIM-BELARE, [17] Picciotto et al. (1968), [18] Mosley-Thompson et al. (1995), [19] Mosley-Thompson et al. (1999), [20] Anschütz et al. (2011).

We discard 66 observations beginning before 1979 and spanning less than eight years. We also discard 12 observations for which the four surrounding grid cells fall in ocean, and seven observations located at specific topographic features for which none of the four surrounding grid cell has an elevation difference less than 200 m with respect to the actual location. After this, we retain 559 model-observation comparisons.

## 3 Results

### 3.1 Evaluation of the modelled SMB

The large spatial Antarctic SMB gradients, shown in Fig. 1a as modelled by MAR forced by ERA-Interim for the period 1979-2015, coincide with a strong interannual variability (Fig. 1b), expressed by a standard deviation of ∼22% of the mean SMB on average over the ice sheet (Fig. 1c). MAR SMB shows no systematic spatial bias (Fig. 1d), with a mean bias of 6 kg m$^{-2}$ yr$^{-1}$ (4% of the mean observed SMB), as well as a very strong correlation with the observed SMB (R$^2$=0.83, p-value<0.01, computed on the logarithm of SMB values, as SMB distributions are log-normal). RACMO2 shows similar performance (mean bias of -3 kg m$^{-2}$ yr$^{-1}$, R$^2$=0.86, computed on the logarithm of SMB as well).

The model-observation comparison by sectors (Fig. 2) reveals a good representation of the coast-to-plateau SMB gradients by both RCMs. MAR and RACMO2 are in good agreement despite MAR not including drifting snow processes whereas RACMO2 does, except in Ross–Marie Byrd Land and in Victoria Land where MAR simulates larger SMB than RACMO2.

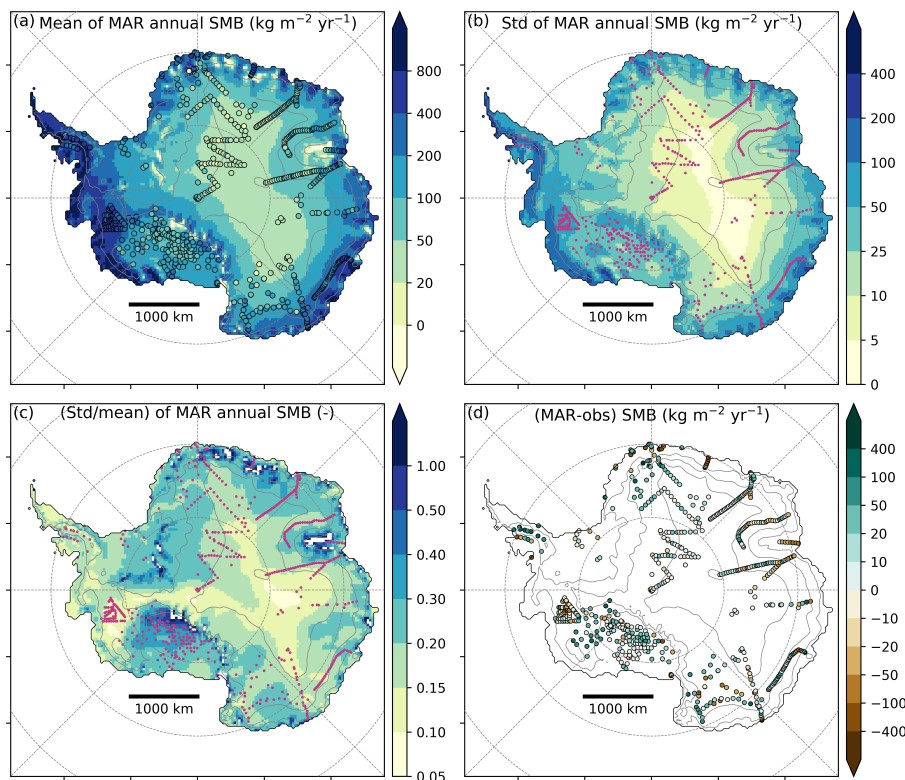

**Figure 1.** MAR SMB for the period 1979-2015: (a) mean annual SMB, with coloured dots showing the observed SMB values (shared colour scale); (b) standard deviation of annual SMB; (c) standard deviation divided by mean annual SMB; (d) difference between MAR and observed SMB on MAR grid cells, following the methodology detailed in Section 2.2.2. Magenta dots in panels b) and c) show the location of SMB observations. Solid grey lines are contours of surface height every 1000 m above sea level. Latitude circles are -60°S, -70°S and -80°S, and longitude lines are from 145°W to 145°E by step of 45°.

25    Another noticeable result is that MAR forced by ERA-Interim, JRA-55 and MERRA2 give very similar results for the SMB spatial pattern, not only at the observation locations (Fig. 2) but also at the ice sheet scale (comparisons of MAR SMB for different forcing reanalyses are shown in Fig. S4, with colormap scales 10 time smaller than in Fig. S5 where MAR is compared to RACMO2). This is why we focus on MAR forced by ERA-Interim in the following.

        We find no significant differences in the SMB simulated by MAR and RACMO2 when integrated over the ice sheet or its
30    major basins (Table 2). SMB is driven by snowfall amounts, which are more than 10 times larger than other SMB components. Snow sublimation in RACMO2 is the sum of sublimation at the surface of the snowpack and of drifting snow sublimation, and is approximately 50 % larger than in MAR which only includes surface snow sublimation. However, surface snow sublimation alone is almost two times larger in MAR than in RACMO2 (Table 2 and spatial patterns shown in Fig. S6), which we investigate in the next section. Modelled surface melt is less than half of the sublimation amount, however liquid water almost entirely refreezes into the snowpack in both models (maps of MAR and RACMO2 modelled melt amounts are shown in Fig. S7).

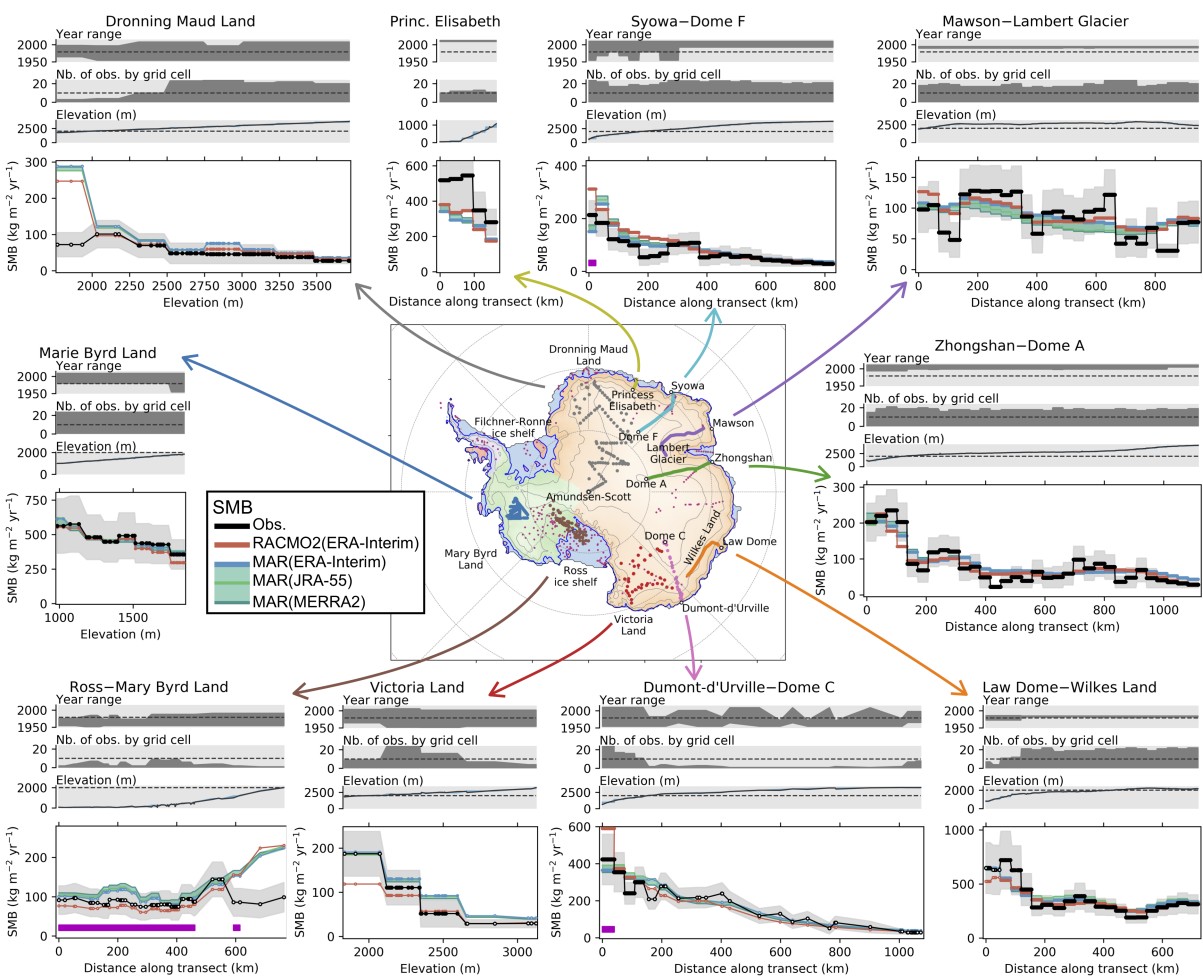

**Figure 2.** Modelled vs. observed SMB for sectors and transects as detailed in Table 1. RACMO2 outputs are bi-linearly interpolated to the MAR grid. SMB values are first averaged on MAR grid cells (Sec. 2.2.2) then along chosen grid direction (Fig. S2) or by elevation bins. Distance along transect starts at the coast. Uncertainty of observed SMB (grey shaded area) is the standard deviation of observations contained in each grid cell (sub-grid variability), estimated as a function of the mean observed SMB (see Fig. S3). Despite SMB values corresponding to grid cell averages, we display one marker for each observation, with the *x* axis corresponding to the observation location along transect or elevation. For observed SMB plots, markers with white faces are for bins containing less than 10 observations and black faces for bins containing more than 10 observations. Magenta bands mark grid cells where more than 15 % of precipitation sublimates in the katabatic layers according to Grazioli et al. (2017). The map shows the main Antarctic basins: Antarctic Peninsula in purple, West Antarctic ice sheet in green, and East Antarctic ice sheet in orange. Ice shelves are mapped in blue, grounded islands in red, and the blue line shows the location of the grounding line.

Temporal variability of the SMB and its components is fully driven in both RCMs by the forcing reanalyses and are therefore

**Table 2.** Antarctic integrated SMB on average for 1979-2015 $\pm$ one standard deviation of annual values, in $\mathrm{Gt\ yr}^{-1}$. Antarctic Ice Sheet (AIS) and basins geometry are based on Rignot basins (Shepherd et al., 2018), shown in Fig. 2. RACMO2 is bi-linearly interpolated on MAR grid and the same mask is applied to both models, with area given for this mask. SMB is computed as follows: MAR SMB = Snowfall + Rainfall − Surface snow sublimation − Run-off; RACMO2 SMB = Snowfall + Rainfall - Surface snow sublimation - Drifting snow sublimation - Drifting snow transport - Run-off.

| Basin | Area ($10^6$ km$^2$) | Component (Gt yr$^{-1}$) | MAR(ERA-Interim) | RACMO2(ERA-Interim) |
|---|---|---|---|---|
| Total AIS w/o Peninsula | 13.41 | SMB | $2200 \pm 115$ | $2177 \pm 107$ |
| | | Snowfall | $2306 \pm 111$ | $2339 \pm 107$ |
| | | Rainfall | $6 \pm 1$ | $2 \pm 1$ |
| | | Surface snow sublimation | $111 \pm 10$ | $57 \pm 4$ |
| | | Drifting snow sublimation | – | $101 \pm 5$ |
| | | Drifting snow transport | – | $5 \pm 0$ |
| | | Run-off | $1 \pm 1$ | $1 \pm 1$ |
| | | Melt | $40 \pm 20$ | $68 \pm 30$ |
| Total AIS | 13.83 | SMB | $2517 \pm 111$ | $2516 \pm 105$ |
| Grounded AIS w/o Peninsula | 12.04 | SMB | $1923 \pm 100$ | $1857 \pm 94$ |
| | | Snowfall | $1995 \pm 97$ | $1987 \pm 94$ |
| | | Surface snow sublimation | $77 \pm 8$ | $39 \pm 3$ |
| | | Drifting snow sublimation | – | $87 \pm 4$ |
| Grounded AIS | 12.27 | SMB | $2120 \pm 99$ | $2068 \pm 93$ |
| Grounded East AIS | 9.77 | SMB | $1170 \pm 89$ | $1121 \pm 80$ |
| | | Snowfall | $1245 \pm 87$ | $1225 \pm 82$ |
| | | Surface snow sublimation | $77 \pm 6$ | $34 \pm 3$ |
| | | Drifting snow sublimation | – | $66 \pm 4$ |
| Grounded West AIS | 2.11 | SMB | $675 \pm 62$ | $643 \pm 62$ |
| | | Snowfall | $675 \pm 61$ | $668 \pm 62$ |
| | | Surface snow sublimation | $1 \pm 3$ | $4 \pm 1$ |
| | | Drifting snow sublimation | – | $20 \pm 2$ |
| Grounded Islands | 0.16 | SMB | $78 \pm 7$ | $93 \pm 8$ |
| Grounded Peninsula | 0.23 | SMB | $198 \pm 26$ | $211 \pm 27$ |

strongly correlated with each other (time series shown in Fig. S8). We do not elaborate on the SMB temporal variability here as this aspect will be further detailed in a forthcoming study.

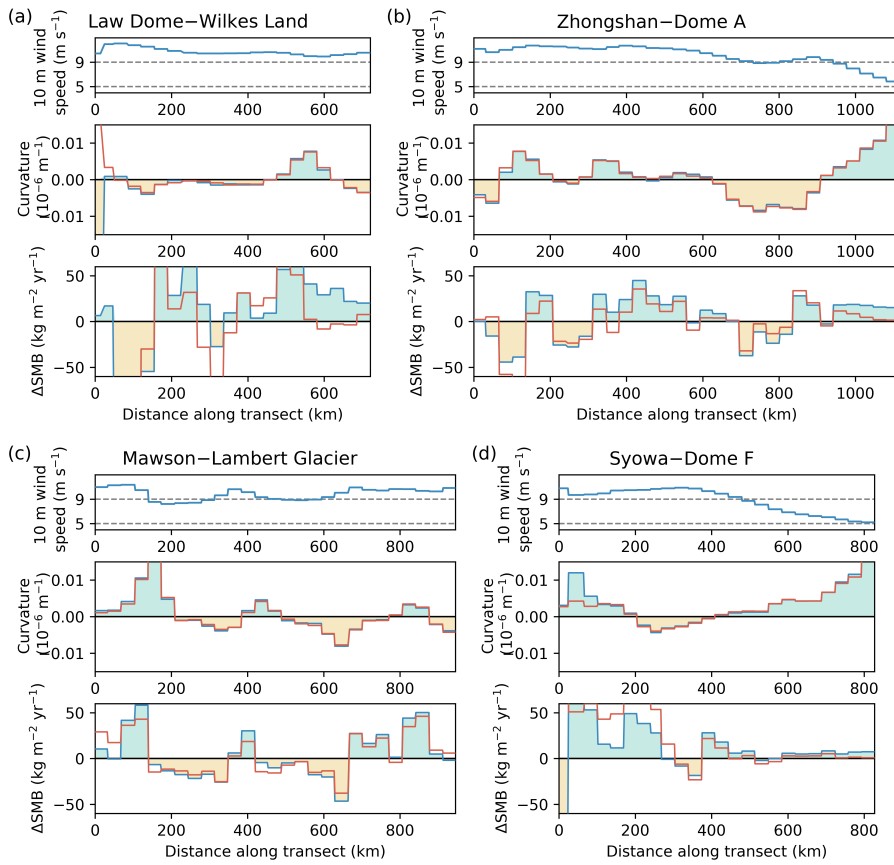

**Figure 3.** For each transect, we show (top) annual mean 10 m wind speed, (middle) curvature of elevation and (bottom) modelled SMB minus observed SMB. Blue lines and colour shading are for MAR(ERA-Interim) outputs and red lines are for RACMO2(ERA-Interim) outputs. Values are computed as in Fig. 2. For Law Dome–Wilkes Land, MAR SMB is shifted by $-30 \ \mathrm{kg \ m^{-2} \ yr^{-1}}$.

## 3.2 Drifting snow transport features

Local fluctuations of the observed SMB around the smooth modelled SMB gradients are apparent along the four stake transects covering more than 500 km: Law Dome–Wilkes Land, Zhongshan–Dome A, Mawson–Lambert Glacier, and Syowa–Dome F. We related these fluctuations to drifting snow transport. Indeed, the snow eroded from the snowpack is loaded into the atmosphere, where it can sublimate and be transported by the wind. Katabatic winds blowing on the surface of the ice sheet result from the downslope gravity flow of cold, dense air. As a consequence, the surface wind divergence, which drives the snowdrift mass transport, is strongly related to the curvature of the topography, and both have similar spatial patterns (shown in Fig. S9). This is because slopes becoming steeper (crests, positive curvature) will lead to wind speed acceleration (positive wind divergence), thus to drifting snow export (mass loss), whereas slopes becoming more gentle (valleys, negative curvature) will lead to wind speed deceleration (negative wind divergence), thus to drifting snow deposit (mass gain).

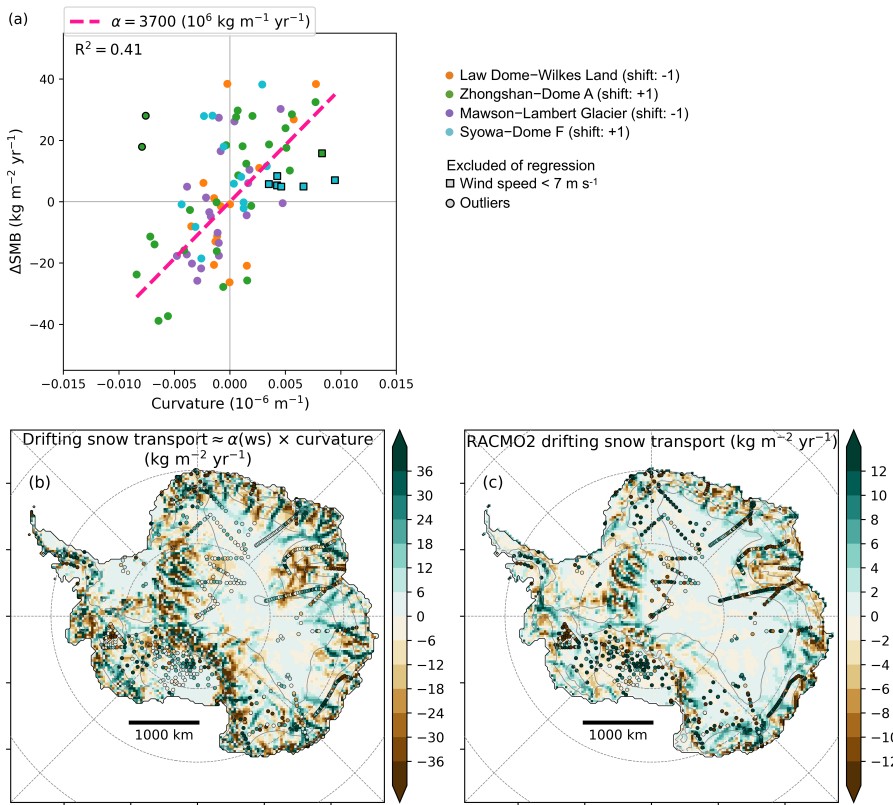

**Figure 4.** (a) Difference in SMB by grid cell ($\Delta$SMB) between MAR(ERA-Interim) and observations for four transects (Law Dome–Wilkes Land, Zhongshan–Dome A, Mawson–Lambert Glacier, and Syowa–Dome F) vs. surface curvature on MAR grid. Curvature is shifted by $\pm$ 1 grid cell according to the maximum correlation with $\Delta$SMB (Fig. S10). Linear regression through the origin is plotted with a dashed pink line. We excluded of regression two outliers (dots with black outlines) and seven data for which MAR annual 10 m wind speed is lower than 7 m s$^{-1}$ (squares with black outlines). (b) Estimate of mean annual drifting snow transport based on a scaling of the curvature: drifting snow transport (kg m$^{-2}$ yr$^{-1}$) = $\alpha$ ($10^6$ kg m$^{-1}$ yr$^{-1}$) $\times$ curvature ($10^{-6}$ m$^{-1}$), with $\alpha = 0$ kg m$^{-1}$ yr$^{-1}$ for wind speed lower than 5 m s$^{-1}$, $\alpha = 3700 \ 10^6$ kg m$^{-1}$ yr$^{-1}$ for wind speed greater than 9 m s$^{-1}$, and $\alpha$ linearly increasing as a function of wind speed in between, around the 7 m s$^{-1}$ wind speed threshold. Wind speed is the annual mean of 10 m wind speed modelled by MAR(ERA-Interim). Coloured dots show the difference between MAR SMB and observed SMB with the same colour scale. (c) Mean annual drifting snow transport flux in RACMO2 on average for 1979-2015 (kg m$^{-2}$ yr$^{-1}$). Coloured dots show the difference between MAR SMB and observed SMB with the same colour scale.

To test our hypothesis, we computed the mean curvature of the MAR 35$\times$35 km elevation field. In Fig. 3, we notice that both RCMs commonly exhibit an excess of accumulation on crests and a deficit of accumulation in valleys, in the range of $\pm$40 kg m$^{-2}$ yr$^{-1}$. To quantify this curvature effect, we correlate MAR SMB bias ($\Delta$SMB) with the curvature. For each transect, we apply a constant shift of $\pm$ one grid cell to the curvature in order to find the maximum correlation with $\Delta$SMB. For three out of the four transects, we find only one shift for which the correlation is significant, and for remaining transect (Syowa–Dome

F) we find no significant correlation (Fig. S10). The sign and the amplitude of those shifts are in line with curvature being used as a proxy for wind divergence, as they are consistent with the Coriolis wind deflection westward of the topography gradient (detailed in Fig. S11). After applying those shifts, we find that the difference between modelled and observed SMB (in kg m$^{-2}$ yr$^{-1}$) is scaled to approximately $3700 \pm 1100$ (in $10^6$ kg m$^{-1}$ yr$^{-1}$) times the curvature (in $10^{-6}$ m$^{-1}$), with a significant relationship ($R^2 = 0.41$, Fig. 4a), when the mean annual 10 m wind speed ($ws_{10}$) is greater than seven m s$^{-1}$. For lower wind speed ($ws_{10} < 7$ m s$^{-1}$), we no longer observe any relationship between model bias in SMB and curvature (horizontally aligned squares in Fig. 4a). This is consistent with the drifting snow transport process which requires the wind speed to reach threshold values for the erosion to be initiated (Amory et al., 2015).

Hence, a large part of the discrepancies between modelled and observed SMB is explained by surface curvature when wind speed is sufficiently high, which we relate to the unresolved drifting snow transport in MAR. We are able to catch the drifting snow transport signal because drifting snow sublimation is negligible for the four studied transects, as they are located at high elevation, upper than 2000 m above sea level (a.s.l.), where the cold atmosphere has low capacity to be loaded with moisture (see detailed analysis in Fig. S12). The moisture holding capacity of the atmospheric boundary layer is an upper bound for drifting snow sublimation and quickly tends to zero when the mean air temperature decreases below -30°C, which is the case along most of the transects, whereas the amplitude of observed SMB fluctuations around the smooth SMB gradient is independent of the temperature (Fig. S13).

Consequently, we propose that drifting snow transport fluxes ($ds_{tr}$) not resolved by MAR can be estimated as a scaling of curvature depending of wind speed: $ds_{tr} = \alpha(ws_{10}) \cdot curvature$ (Figure 4b). The scaling factor $\alpha(ws_{10})$ depends on wind thresholds to simulate the transition between no drifting snow transport for low wind speed ($\alpha = 0$ for $ws_{10} < 5$ m s$^{-1}$) and drifting snow transport scaled to curvature for high wind speed ($\alpha = 3700 \ 10^6$ kg m$^{-1}$ yr$^{-1}$ for $ws_{10} > 9$ m s$^{-1}$), with a linearly increasing scaling factor between 5 and 9 m s$^{-1}$ for a smooth transition around the 7 m s$^{-1}$ wind threshold defined above. That estimate of drifting snow transport fluxes shows little sensitivity to the choice of the wind thresholds and of the scaling factor (see fluxes summed over the ice sheet for different thresholds and scaling factors in Table S2). The spatial pattern of drifting snow transport we obtain is comparable to the one simulated by RACMO2 (Fig. 4c), except that it gives fluxes more than three times larger than in RACMO2 (Table S2, and note the different colour map scales between Fig. 4b and 4c). The drifting snow transport estimate consists in a redistribution of mass with negligible net mass loss over the Antarctic ice sheet (total AIS mass gain of ~75 Gt yr$^{-1}$ and total AIS mass loss of ~80 Gt yr$^{-1}$, see Table S2).

Our drifting snow transport estimate gives a good constraint for drifting snow fluxes above 2000 m a.s.l., where low temperatures induce negligible atmospheric sublimation. As drifting snow transport is proportional to the amount of snow in suspension in the atmosphere, quantifying this flux also enables to constrain the amount of snow eroded from the snowpack to the atmosphere, which drives drifting snow sublimation fluxes at lower elevation. This is of importance as drifting snow sublimation is a much larger mass sink than drifting snow transport over the whole ice sheet (Palm et al., 2017; Lenaerts et al., 2012a) but is still poorly constrained because observations are very scarce bellow 2000 m a.s.l. where it occurs.

Drifting snow sublimation included in RACMO2 and not in MAR moisten the surface atmospheric layers, consequently reducing the sublimation at the surface of the snowpack. This might explains the stronger surface snow sublimation in MAR

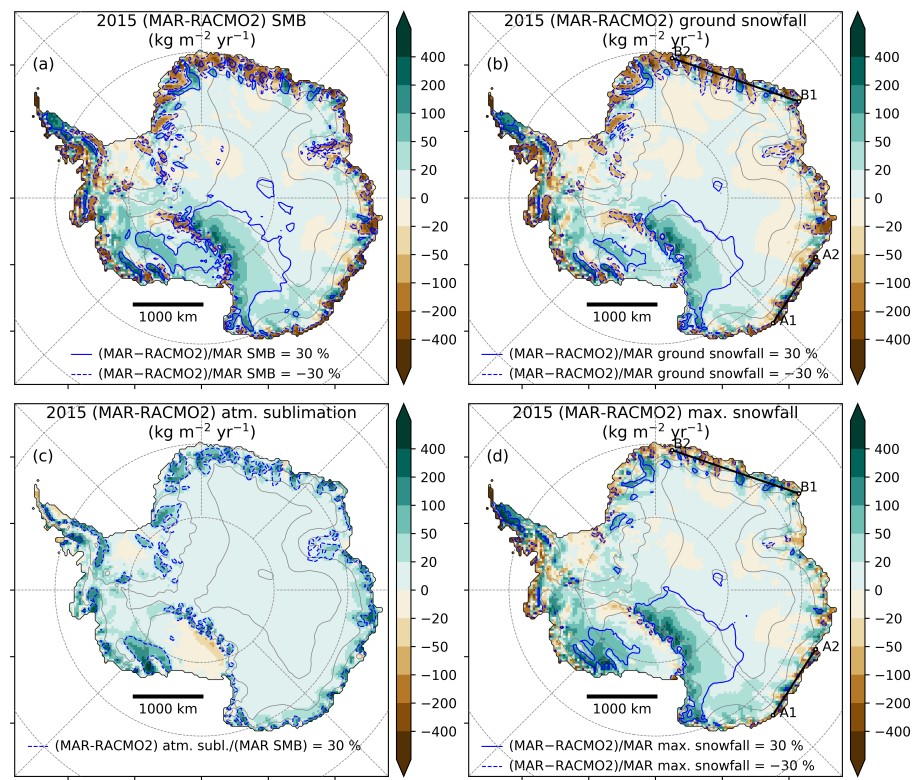

**Figure 5.** The four maps show mass fluxes in kg m$^{-2}$ yr$^{-1}$ for the year 2015. (a) Difference in SMB between MAR and RACMO2. Blue lines delimitate areas where the SMB difference is 30 % greater than MAR SMB, with solid lines when MAR is greater than RACMO2 and dashed lines when MAR is lower than RACMO2. (b) Same as a) but for the snowfall amounts at the ground. (c) Same as a) but for the sublimation of precipitation in the atmospheric layers. (d) Same as a) but for the maximum snowfall amount (equal to ground snowfall plus atmospheric sublimation). Locations of transects A1-A2 and B1-B2 extracted in Fig. 6 are shown in panels b) and d).

than in RACMO2 (Table 2 and Fig. S6). However, drifting snow sublimation is a potentially larger mass sink than surface
5   snow sublimation, as drifting snow particles are continuously ventilated and fully exposed to the ambient air. Consequently, by accounting for drifting snow in MAR we expect that the drifting snow sublimation mass sink could be enhanced at the expense of surface snow sublimation at the ice sheet margins.

### 3.3   Sublimation of precipitation in low-level atmosphere

As described above, MAR and RACMO2 regional climate models forced with ERA-Interim simulate similar spatial patterns
10   for SMB as compared to observations (Fig. 2). However, at the ice sheet scale, MAR and RACMO2 SMB show regional discrepancies (shown in Fig. 5a for 2015, and similar than the 1979-2015 mean shown in Fig. S5a) which are primarily the result of differences in simulated snowfall rates (Fig. 5b, and  S5b). We notice that areas where MAR snowfall is much lower than RACMO2 snowfall (Fig. 5b, dashed blue lines) coincide almost exactly with the pattern of precipitation that is able to

sublimate in the low-level atmosphere according to Grazioli et al. (2017). In that study, the amount of atmospheric sublimation is quantified for the year 2015 using atmospheric modelling constrained with precipitation radar observations. Atmospheric sublimation happens because the katabatic surface air flux, moving from high-elevated inland plateau toward sea level, is subject to adiabatic compression when it moves downslope. This compression induces an increase in air temperature which reduces relative humidity and drives sublimation rates in the lower troposphere ($\sim$first 1000 m above the ground), enhanced in the katabatic channels at the ice sheet margins.

To deepen this analysis, we re-ran MAR for the year 2015 in order to save the full atmosphere snowfall fields. From the daily 3D snowfall amounts, we derived the atmospheric sublimation amount from the difference between the maximum snowfall and the ground snowfall in each atmospheric column, as in Grazioli et al. (2017). The same was done for RACMO2. We find that the atmospheric sublimation simulated by MAR (363 Gt for the year 2015 over the grounded ice sheet) is higher than estimated in Grazioli et al. (2017) (299 Gt after interpolation on the same mask), and much higher than simulated by RACMO2 (128 Gt, Fig. 5c). A major difference between MAR and RACMO2 is the advection of precipitation in the atmosphere: in MAR, precipitating particles are explicitly advected through the atmospheric layers until they reach the surface, while in RACMO2, precipitation is added to the surface without horizontal advection, and is able to interact with the atmosphere only in a single time step (6 min in this simulation). Consequently, atmospheric sublimation is likely to be underestimated in RACMO2.

We conclude, in agreement with Grazioli et al. (2017), that atmospheric sublimation is a major mass sink at the ice sheet margins in MAR, as for the year 2015 it represents 16 % of the snowfall loaded on the grounded ice sheet (12 % in Grazioli et al., 2017), and 26 % for areas bellow 1000 m a.s.l. (17 % in Grazioli et al., 2017).

It is noticeable that very few SMB observations are available in areas where Grazioli et al. (2017) identify low-level sublimation, marked by magenta bands in Fig. 2. Except for Ross–Marie Byrd Land, the only other areas where low-level sublimation is greater than 15 % of the total precipitation as defined by Grazioli et al. (2017) are close to Dumont d'Urville (coastal Adelie Land) and to Syowa (coastal Dronning Maud Land). In those areas the SMB amount is indeed larger in RACMO2 than in MAR and in observations. Both RCMs overestimate SMB around 2000 m a.s.l. in Dronning Maud Land and in Ross–Marie Byrd Land (Fig. 2), which could indicate katabatic channels not enough resolved by the topography of the models.

### 3.4 Precipitation formation and advection

Differences between MAR and RACMO2 snowfall fields are strongly reduced when considering the maximum snowfall amounts (before sublimation in the low-level atmosphere) rather than the ground snowfall amounts (Fig. 5b and Fig. 5d). However, MAR snowfall rates generally exceed those simulated by RACMO2, by more than 30 % on the lee side of the West AIS (Marie Byrd Land toward Ross ice shelf), on the lee side of the Transantarctic Mountains (Victoria Land) and close to crests at the ice sheet margins. MAR maximum snowfall rates are lower than simulated by RACMO2 windward of topographic barriers and in valleys at the ice sheet margins. This spatial pattern looks similar to the one obtained in RACMO2 when delaying the conversion of cloud ice/water into snow/rain (Fig. 3a of van Wessem et al., 2018). This change led to both ice and water clouds lasting longer in the atmosphere before precipitating and therefore being advected further towards the ice sheet interior (van Wessem et al., 2018).

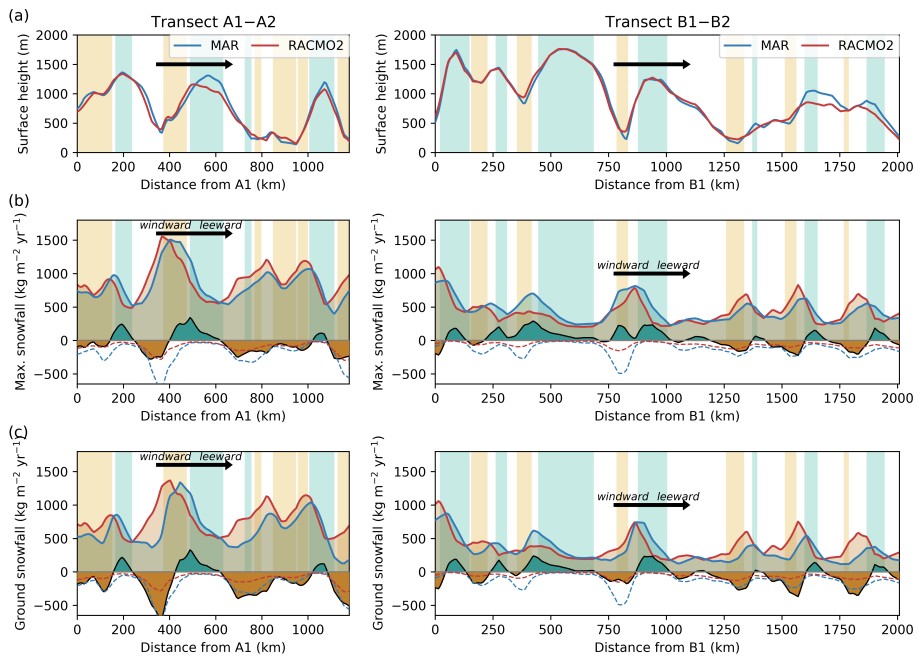

**Figure 6.** MAR and RACMO2 simulated fields for the year 2015, extracted with a bi-linear interpolation for (left) transect A1-A1 and (right) transect B1-B2 (locations shown in Fig. 5b and Fig. 5d). Each panel shows MAR fields (blue lines) and RACMO2 fields (red lines) for (a) surface height, in m a.s.l.; (b) maximum snowfall amounts, equal to ground snowfall plus atmospheric sublimation, in $\mathrm{kg\ m^{-2}\ yr^{-1}}$; and (c) snowfall amounts at the ground, in $\mathrm{kg\ m^{-2}\ yr^{-1}}$. In (b) and (c), the thick black line is for the difference in snowfall between MAR and RACMO2 (MAR-RACMO2), with green-filled areas when MAR snowfall is larger than RACMO2 snowfall, and brown-filled areas when MAR snowfall is lower than RACMO2 snowfall (same convention as in Fig. 5); the dotted lines are for the atmospheric sublimation modelled by MAR (blue) and by RACMO2 (red), negative when it induces a decrease in precipitation; light coloured bands show crests (light blue, curvature of MAR topography greater than $0.005\ 10^{-6}\ \mathrm{m^{-1}}$) and valleys (light yellow, curvature of MAR topography lower than $-0.005$ $10^{-6}\ \mathrm{m^{-1}}$). The thick black arrows show the main 800 hPa wind direction during cyclonic activity.

For a more in-depth analysis, we extract MAR and RACMO2 snowfall rates on two transects at the ice sheet margins (Fig. 6),
5  following the main wind direction during cyclonic activities (locations shown in Fig. 5b and Fig. 5d). On these transects the observed difference in maximum snowfall between MAR and RACMO2 is largely explained by a phase difference in the snowfall peaks windward of the topographic barriers, with MAR peaking closer to the crests than RACMO2 (Fig. 6b). This induces a wave-like pattern of precipitation difference strongly related to the shape of the topography, with larger snowfall amounts in MAR than in RACMO2 just windward of crests, and lower snowfall amounts in MAR than in RACMO2 around windward valleys. At the ground, lower snowfall in MAR than in RACMO2 in valleys is amplified by low-level atmospheric sublimation which peaks in katabatic channels (Fig. 6c).

Observations do not enable to definitively discriminate one model against the other, but we observe a general tendency for MAR to overestimate accumulation on Ross–Marie Byrd Land and close to ice sheet summits (Dome C, Dome A, Dome F, see

Fig. 1d and Fig. 2). Close to summits the wind is low, so missing drifting snow transport process is unlikely explanation for a positive bias in SMB modelled by MAR (Fig. 4b). Over the Greenland ice sheet, MAR tends to overestimate ice cores based accumulation inland (Fettweis et al., 2017) while RACMO2 underestimates it (Noël et al., 2018).

We conclude that the differences in MAR and RACMO2 snowfall patterns are very likely related to differences in the advection of precipitation inland, which may arise from (i) the different advection of precipitating particles to the ground described in Section 3.3, (ii) different timing of precipitation formation (cloud/precipitation conversion thresholds), and/or (iii) different dynamical response to the topographic forcing, caused either by different dynamical cores or by the different resolutions (the 27 km resolution in RACMO2 better resolves the ice sheet topography than the 35 km resolution in MAR).

## 4    Discussion and conclusion

In our study, we evaluate new estimates of the Antarctic SMB obtained with the polar regional climate model MAR ran for the first time for decades-long simulations at the scale of the whole Antarctic ice sheet. We use model settings comparable to previous MAR simulations over Greenland (Fettweis et al., 2017) but with a specific upper atmosphere relaxation and new surface snow density and roughness length parameterisations. We present simulations of MAR forced by ERA-Interim, JRA-
55 and MERRA2 for the satellite era (1979-2015) where we can rely on reanalyses products. Remarkably, MAR forced by those three reanalyses give similar spatial and temporal SMB patterns. We also compare MAR with the latest simulations of the RCM RACMO2 forced by ERA-Interim (van Wessem et al., 2018). We find no significant differences between MAR and RACMO2 SMB when integrated on the AIS and its major basins (Table 2).

As the dominant feature of the Antarctic SMB is its strong coast-to-plateau gradient, we extract stake transects and sectors
with large elevation ranges from the GLACIOCLIM-SAMBA SMB observational dataset. We show that both RCMs show similar performances when compared to observations, with a good representation of the SMB gradient (Fig. 2). But more importantly, we outline and quantify missing or underestimated processes in both RCMs.

Along stake transects, we relate 100 km-scale fluctuations of observations around the smooth modelled SMB pattern to the shape of the ice sheet captured on the $35 \times 35$ km MAR grid. Both RCMs accumulate too much snow on crests, and not
enough snow in valleys, as a result of drifting snow transport fluxes not included in MAR and probably underestimated in RACMO2 by a factor of three (Fig. 4). In the RACMO2.3p2 version used here, the modified drifting snow routine induced almost halved drifting snow transport and sublimation fluxes compared to the previous RACMO2.3p1 version (Lenaerts and van den Broeke, 2012). In a recent study combining satellite observation of drifting snow events and reanalysis products, Palm et al. (2017) estimate the drifting snow sublimation to be about $\sim$393 Gt yr$^{-1}$ over the Antarctic ice sheet, vs. 181 Gt yr$^{-1}$ in RACMO2.3p1 and 102 Gt yr$^{-1}$ in RACMO2.3p2 (van Wessem et al., 2018). Consequently, observational constraints from our study and from Palm et al. (2017) both tend to confirm that drifting snow transport and sublimation fluxes are likely much larger than previous model-based estimates and need to be (better) resolved and constrained in climate models.

We also point out that MAR generally simulates larger SMB and snowfall amounts than RACMO2 inland, particularly on
the lee side of the Transantarctic Mountains and on crests at the ice sheet margins, whereas MAR simulates lower snowfall than

RACMO2 windward of mountain ranges and promontories. Sublimation of precipitating particles in low-level atmospheric layers is largely responsible for the significantly lower snowfall rates in MAR than in RACMO2 in valleys at the ice sheet margins. As precipitating snow particles have larger time residence in the atmosphere in MAR than in RACMO2 (Section 3.3), amounts of precipitation lost by sublimation in katabatic channels are more than twice as much in MAR as in RACMO2. The remaining

spatial differences in snowfall between MAR and RACMO2 are attributed to differences in advection of precipitation, snowfall particles being likely advected too far inland in MAR.

Atmospheric sublimation represents $429\ \mathrm{Gt\ yr^{-1}}$ in MAR over the whole AIS (Peninsula excluded) for the year 2015, 89 % of which is lost below 2000 m a.s.l., and 61 % below 1000 m a.s.l.. This might be of importance for the mass balance of glacier drainage basins (SMB minus discharge, Rignot et al., 2008; Shepherd et al., 2018), as ice streams are typically channel-shaped

areas affected by low-level sublimation of precipitation. Consequently, we note the importance of saving precipitation fluxes in models at least 1300 m above the ground for comparison with CloudSat products, but ideally at all model levels below 1500 m above the ground to be able to compute sublimation of precipitation in the low-level atmospheric layers. This will become a standard output in forthcoming MAR simulations.

We expect that accounting for drifting snow in MAR will lead to significant improvements in describing the Antarctic SMB

and surface climate, as it will enable (1) a quantification of the drifting snow sublimation mass sink, (2) a more realistic representation of relative humidity and temperature in the boundary layer, and (3) an explicit modelling of the drifting snow transport from crests to valleys. Exploring the impact of horizontal and vertical model resolution on drifting snow estimates and on sublimation of precipitation in katabatic channels will also be of importance as those processes are related to the shape of the ice sheet and to the advection of precipitation in the atmosphere. The accuracy of the topography has to be considered as

well, as digital elevation models are in constant improvement over the Antarctic ice sheet (e.g. Slater et al., 2018) and should be regularly updated in climate models.

*Code and data availability.* Python scripts developed for this study as well as all required data are available at https://gitlab.com/cecileagosta/antarctica-smb-20c.git. The last version of MAR is freely distributed at http://mar.cnrs.fr/. Monthly MARv3.6.4 outputs from this study are freely available at ftp://ftp.climato.be/fettweis/MARv3.6/Antarctic/, together with the associated MAR source code. The ECMWF reanalyse ERA-Interim 6-hourly outputs were downloaded from http://apps.ecmwf.int/datasets/. The MERRA2 reanalyse 6-hourly outputs were downloaded from https://disc.sci.gsfc.nasa.gov/. The JRA-55 reanalyse 6-hourly outputs were downloaded from https://rda.ucar.edu/datasets/dS728.0/.

*Author contributions.* Cécile Agosta set-up the MAR model for Antarctica with several adaptations, performed model simulations and analysed model outputs and observations. Cécile Agosta, Anais Orsi, Xavier Fettweis and Vincent Favier designed the study. Cécile Agosta, Xavier Fettweis, Hubert Gallée, Charles Amory and Christoph Kittel developed the MAR model and contributed to the MAR set-up and output analyses. Xavier Fettweis and Hubert Gallée are the main developer of the MAR model. Michiel R. van den Broeke, J. Melchior van Wessem, Jan T.M. Lenaerts and Willem Jan van de Berg contributed to RACMO2 output analyses. All authors contributed to discussions in
5   writing this manuscript.

*Competing interests.* The authors declare that they have no conflict of interests.

*Acknowledgements.* We thank Kenichi Matsuoka, Massimo Frezzotti and the second anonymous reviewer for their constructive and insightful comments, that led to a much improved paper. Cécile Agosta performed MAR simulations during her Belgian Fund for Scientific Research (F.R.S.-FNRS) research fellowship. Computational resources have been provided by the Consortium des Équipements de Calcul Intensif (CÉCI), funded by the F.R.S.-FNRS under Grant No. 2.5020.11. We acknowledge Jacopo Grazioli for sharing low-level sublimation product and his expertise of this dataset. We acknowledge Yetang Wang for sharing his updated version of the GLACIOCLIM-SAMBA dataset. We acknowledge Christophe Genthon for fruitful discussions and suggestions. The authors acknowledge the support from Agence Nationale de la Recherche scientific for the scientific traverses in Antarctica and the associated research on climate and surface mass balance modeling, projects ANR-14-CE01-0001 (ASUMA) and ANR-16-CE01-0011 (EAIIST).

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
