# Peer review of "Estimation of the Antarctic surface mass balance using the regional climate model MAR (1979-2015) and identification of dominant processes"

_The Cryosphere, 2018_

## Referee Comment (RC1) · M. Frezzotti (Referee) · 23 May 2018

This paper presents comparison between the Surface Mass Balance outputs of two regional climate models MAR and RAMCO2. The paper contributes to on-going debate concerning the estimation of Antarctic SMB and the result of atmospheric model to reproduce SMB. The manuscript subject is appropriate for "The Cryosphere" and the result are very interesting and must be support, however the manuscript must be improved. My main concerns are the following issues: • Snow precipitation is removed in atmosphere by wind driven process that are mainly due to katabatic wind that follow the downslope flow, on the large sloping glaciers in the Antarctica, the Cori-
olis force becomes very significant force. The geological sedimentation process of erosion/deposition cannot be applied to the snow that sublimate when remain in dry atmosphere. Topographic slope and curvature MUST be calculated along the main katabatic wind direction, that due to Coriolis force can be very different to topographic slope used by Authors as curvature (see Frezzotti et al., 2002, 2004, 2007; Scambos et al., 2012, Das et al., 2013, 2015; Palm et al., 2011, 2017). • Authors have supposed that "snow is usually eroded from topographic crests and collected in the valley", this hypothesis is not corroborated from any field observation (see GPR profile of Talos Dome, Frezzotti et al. 2007, Fujita et al. 2011, Stake profile Syowa-Dome F, Zhongshan-Dome A etc.) in East Antarctica. • On the ice divide (crest) the wind are slower and does not eroded/sublimate, whereas in the valley the katabatic wind speed increase and sublimate the drifting/blowing snow. SMB measurements point out that in East Antarctica there are very few evidence of erosion/deposition less than 10%, most of the process are snow erosion/blowing/sublimation without any redeposition (see Frezzotti et al., 2004, 2007; Scarchilli et al., 2010; Minghu et al., 2011, Ding et al., 2017). Erosion/deposition process can occur on saturated contion as ice shelf or where the slope along wind direction does not permit the sublimation because the atmosphere became saturated soon. • It is not explained why the erosion/drifting module of MAR are not used. Drifting/sublimation snow is very important component of SMB, as also reported by authors. • Authors point out that at regional and continental scale the results of the simulated SMB do not present significant difference and are in good agreement (pag 6), despite significant differences components in the negative value of SMB, in absolute value can be correct, but the comparison of the single SMB components are very different. • Authors must be taking in account the coarse resolution used, in particular in the coastal and confluence area where 35 km of horizontal resolution are too coarse to simulate valley, this influence strongly the wind speed and relative sublimation process. • Due to the different climatic condition, mainly melt and limited katabatic wind phenomena, the SMB components analysis of the Peninsula, West Antarctica and Ross/Filchner-Ronne ice shelves area should be

analysed separately by EAIS.

Detail: Pag 2 line 6, also MB from GRACE or altimeter use extensively SMB estimation. Pag 3 line 28, "Fresh snow" density cannot be 400 kg m3, use "surface snow" Pag. 4 line 12, drifting snow is not a negligible components, and cannot compensate by higher surface sublimation, result from MAR drifting snow should be presented. Pag 6 line 24 76 kg/m2/yr is not a negligible value and represent about 60%!!! Please comment and integrating. Pag 6-7-8-9-10 see above main comments Pag 9 table 2 The different component of SMB must be tabled in different way, positive component: snowfall and rainfall; negative term: sublimation and run off; surface process: melt-refreezed into the snowpack and erosion-deposition. Pag 10 line 8-12 erosion-deposition is a "sedimentation" phenomenon, if snow sublimate and then redeposit under snowfall it is not exported in atmosphere/ocean, rewriting the text. Pag 11 line 5 I do not understand, MAR drifting module is used or not, why several repetition about MAR drifting module? Pag 12-13 The Grazioli paper is very interesting, but snowfall generally occurs under cyclonic storm and not under "pure" katabatic wind phenomena. Katabatic wind arrives later with strong blowing snow phenomena and related sublimation (see Palm et al., 2011, 2017; Scarchilli et al., 2010). Wind during cyclonic storm are variables and not from dry high-elevated inland plateau toward sea level. This does not exclude that wind sublimation occur during a storm, but normally during marine storm the atmosphere is already saturated with low capacity of sublimation. Pag 14 line 3-7 Wind crust area reported in Scambos et al. 2012 are related to hiatus in accumulation driven by sublimation wind process, it is not clear the relation with observed difference between MAR and RACMO2 snowfall. The wind crust is the extreme phenomena where the ratio between snowfall and wind sublimation conduct to hiatus in accumulation from several to thousand year (see Frezzotti et al., 2002, 2005). Due to the limit of method of Scambos et al., 2012, wind crust are surveyed only in the inland plateau (above 1500 m) where the coarse resolution of models have less impact on the slope along wind direction and therefore wind speed. Wind crust is the upper limit of hiatus, before became blue ice area, but they represent only a limited area of wind drive sublimation

area (see Palm et al. 2011, Frezzotti et al., 2007; Minghu et al., 2011) those are more extended then permanent wind crust surface mapped by Scambos et al., 2011. Models firstly must be reproduce the wind crust hiatus, if they can be representative of the negative term of SMB.

If it could be useful, the SMB Talos Dome transect published on Frezzotti et al., 2007 paper is available for the comparison of models.

Reference: Das, I., Bell, R. E., Scambos, T. A., Wolovick, M., Creyts, T. T., Studinger, M., ... & Van Den Broeke, M. R. (2013). Influence of persistent wind scour on the surface mass balance of Antarctica. Nature Geoscience, 6(5), 367. Das, I., Scambos, T. A., Koenig, L. S., Broeke, M. R., & Lenaerts, J. (2015). Extreme windřice interaction over Recovery Ice Stream, East Antarctica. Geophysical Research Letters, 42(19), 8064-8071. Ding, M., Zhang, T., Xiao, C., Li, C., Jin, B., Bian, L., ... & Qin, D. (2017). Snowdrift effect on snow deposition: Insights from a comparison of a snow pit profile and meteorological observations in east Antarctica. Science China Earth Sciences, 60(4), 672-685. Fujita, S., Holmlund, P., Brown, I., Enomoto, H., Fujii, Y., Fujita, K., ... & Hoshina, Y. (2011). Spatial and temporal variability of snow accumulation rate on the East Antarctic ice divide between Dome Fuji and EPICA DML. The Cryosphere, 5(4), 1057. Frezzotti, Massimo, et al. "Snow dunes and glazed surfaces in Antarctica: new field and remote-sensing data." Annals of Glaciology 34 (2002): 81-88. Frezzotti, M., Pourchet, M., Flora, O., Gandolfi, S., Gay, M., Urbini, S., ... & Severi, M. (2004). New estimations of precipitation and surface sublimation in East Antarctica from snow accumulation measurements. Climate Dynamics, 23(7-8), 803-813. Frezzotti, M., Pourchet, M., Flora, O., Gandolfi, S., Gay, M., Urbini, S., ... & Severi, M. (2005). Spatial and temporal variability of snow accumulation in East Antarctica from traverse data. Journal of Glaciology, 51(172), 113-124. Frezzotti, M., Urbini, S., Proposito, M., Scarchilli, C., & Gandolfi, S. (2007). Spatial and temporal variability of surface mass balance near Talos Dome, East Antarctica. Journal of Geophysical Research: Earth Surface, 112(F2). Minghu, D., Cunde, X., Yuansheng, L., Jiawen, R., Shugui, H., Bo, J., &

Bo, S. (2011). Spatial variability of surface mass balance along a traverse route from Zhongshan station to Dome A, Antarctica. Journal of Glaciology, 57(204), 658-666. Scambos, T. A., Frezzotti, M., Haran, T., Bohlander, J., Lenaerts, J. T. M., Van Den Broeke, M. R., ... & Neumann, T. (2012). Extent of low-accumulation'wind glaze'areas on the East Antarctic plateau: implications for continental ice mass balance. Journal of glaciology, 58(210), 633-647. Palm, S. P., Yang, Y., Spinhirne, J. D., & Marshak, A. (2011). Satellite remote sensing of blowing snow properties over Antarctica. Journal of Geophysical Research: Atmospheres, 116(D16). Palm, S. P., Kayetha, V., Yang, Y., & Pauly, R. (2017). Blowing snow sublimation and transport over Antarctica from 11 years of CALIPSO observations. The Cryosphere, 11(6), 2555. Scarchilli, C., Frezzotti, M., Grigioni, P., De Silvestri, L., Agnoletto, L., & Dolci, S. (2010). Extraordinary blowing snow transport events in East Antarctica. Climate Dynamics, 34(7-8), 1195-1206.
* * *

---

## Referee Comment (RC2) · Anonymous Referee #2 · 26 Jun 2018

Remarks to the Authors

Review of "Estimation of the Antarctic surface mass balance using MAR (1979-2015) and identification of dominant processes" by Cécile Agosta et al.

The Cryosphere Discuss. Manuscript Number: tc-2018-76
* * *
General comments:

This paper presents performance of the polar regional climate model MAR applied in the entire Antarctic Ice Sheet (AIS) for the first time. MAR has been applied and val-

idated in the Greenland ice sheet (GrIS) for a long time, and it is widely recognized as a useful and reliable tool to understand the polar climate system. In the present study, the authors follow basically the same MAR model configuration developed in the GrIS. In addition, they decrease horizontal and vertical resolutions (due to the AIS's much larger area than the GrIS), use a boundary relaxation of upper air temperature and wind speed, employ an optimized fresh snow density parameterization for the AIS, and utilize a dynamic parameterization for the aerodynamic roughness length. This reviewer finds that these modifications are reasonable to conduct this kind of study. The model forced by the European Centre for Medium-Range Weather Forecasts (ECMWF) Interim reanalysis (ERA-Interim) is evaluated in terms of surface mass balance (SMB) using the in-situ data obtained during 1979-2015. In this process, the authors also refer to model simulation results by another polar regional climate model known as RACMO2 (horizontal resolution is 27 km) forced by the same reanalysis data to identify important physical processes that influences the AIS SMB simulations. The authors find that both models tend to accumulate too much snow on crests, whereas not enough snow in valleys. Here, the authors attribute the main reason for this discrepancy to the insufficiency of drifting snow-induced erosion-deposition process modeling in both models. When calculated SMBs by MAR and RACMO2 are integrated over the AIS, no significant differences are found between these two results; however, geographical SMB patterns for both models differ significantly, which suggest that there are many things to do to develop a truly reliable polar regional climate model for the AIS. In valleys, RACMO2-simulated precipitation is larger than that by MAR: it is mainly attributed to a difference in modeling approach for sublimation in unsaturated katabatic layer. On the other hand, larger precipitation in the inland AIS is simulated by MAR, because of the difference in horizontal resolution set in both models, which significantly affect orographic impacts on the simulated precipitation rates.

Overall, this paper is well written and can be informative for readers who are interested in the AIS climate system; however, this reviewer thinks that some discussions are not deepened sufficiently and suggests the following points to be considered before the

publication. In the following part, this reviewer gives specific comments. Page and line numbers are denoted by "P" and "L", respectively.

————————————————————————————————————————————————

Specific comments (major)

P. 9, L. 3: What do the authors mean by "shift" mentioned here? Why is this procedure needed here? Please explain more about the procedure.

P. 11, L. 1.: The authors mention that near surface atmosphere is simulated to be drier in MAR compared to RACMO2. How large is the difference? Please quantify and discuss why the difference was made.

Sect. 3.3: Using MAR, can the authors perform a model sensitivity test where sublimation in unsaturated katabatic layer is not allowed? If results from this sensitivity test are provided, the argument by the authors in this section would become more convincing.

Sects. 3.3 and 3.4: In Sect. 3.4, the authors point out the importance of orographic effects on the precipitation simulations in areas centered on crests. It is interesting the authors don't mention orographic effects on the precipitation simulations at valleys (Sect. 3.3). Do the authors think that considering the process for the low-level sublimation in unsaturated atmosphere (at especially valleys) is more important than setting a higher horizontal resolution to obtain realistic SMB at valleys by a model?

P. 13, L. 33: Can the authors perform a MAR model sensitivity test where the horizontal resolution is set to be 27 km (same as RACMO2) or higher? I know it is computationally demanding, but, results from such a sensitivity test for even only several years would be informative for readers.

————————————————————————————————————————————————

Specific comments (minor)

P. 2, L. 4: Regarding the "several approaches", please list up and explain these approaches briefly here. I believe the information are very informative for readers.

P. 3, L. 18: Why did the authors set the horizontal resolution to be 35 km for MAR in the present study? To perform detailed and solid comparisons between MAR and RACMO2, setting the same horizontal resolution is very ideal.

P. 5, L. 10: Figure 1 basically presents simulation results from MAR, therefore, referring Fig. 1 in this sentence is a bit strange (MAR simulation results don't reproduce the reality, although I agree it certainly does a good job.).

P. 6, L. 8 ∼ 10: I could not follow the explanation here. Could you please detail more?

P. 6, L. 23: For me, it is not easy to understand the authors' intension regarding "oscillates" mentioned here. Could you please reformulate it?

P. 6, L. 23 ∼ 24: In Sect. 3, the authors present the performance of modeled SMB by MAR. They also perform detailed comparisons between simulation results from MAR and RACMO2. In this context, I think it is better to denote the performance of RACMO2 in terms of SMB here in the same manner as MAR (please indicate mean bias and RMSE for RACMO2).

P. 7, L. 10: It is not easy to understand the meaning of "oscillations" mentioned here. Could you please rephrase it?

P. 13, L. 4 ∼ 14: Do the authors mean that the MAR-simulated precipitation at valleys is more realistic compared to the RACMO2-simulated precipitation at valleys? Please describe more clearly.

P. 14, L. 3: "wind glaze area": Please detail more about its definition here.

––––––––––––––––––––––––––––––––––––––––––––––––––––––––––––––––––––

Technical corrections:

Figure 1: Please explain red circles in Figs. 1a to 1c in the caption. It is also the case

for Figs. 4b and 4c.

P. 9, L. 5: "wind speed" -> "10 m wind speed"?

P. 13, L. 22 ∼ 23: In Fig 5b, no description on the altitude of the AIS is provided. Please check it again and revise it.

---

## Author Comment (AC1) · 29 Oct 2018

M. Frezzotti (Referee#1) massimo.frezzotti@enea.it

This paper presents comparison between the Surface Mass Balance outputs of two regional climate models MAR and RAMCO2. The paper contributes to on-going debate concerning the estimation of Antarctic SMB and the result of atmospheric model to reproduce SMB. The manuscript subject is appropriate for "The Cryosphere" and the result are very interesting and must be support, however the manuscript must be improved.

Dear Massimo, thank you for your comments and for your useful analysis and suggestions in this review.

My main concerns are the following issues:

- Snow precipitation is removed in atmosphere by wind driven process that are mainly due to katabatic wind that follow the downslope flow, on the large sloping glaciers in the Antarctica, the Coriolis force becomes very significant force. The geological sedimentation process of erosion/deposition cannot be applied to the snow that sublimate when remain in dry atmosphere. Topographic slope and curvature MUST be calculated along the main katabatic wind direction, that due to Coriolis force can be very different to topographic slope used by Authors as curvature (see Frezzotti et al., 2002, 2004, 2007; Scambos et al., 2012, Das et al., 2013, 2015; Palm et al., 2011, 2017)

*Snowdrift transport vs. drifting snow sublimation*

Following your comments, we thought that the term "erosion-deposition" was misleading as erosion is usually interpreted as the removal of snow from the snowpack to the atmosphere, whereas the flux we wanted to describe was the horizontal advection of drifting snow. We changed this term by "drifting snow transport", which is closer to the "geological sedimentation process of erosion/deposition".

In our article we separated the drifting snow transport and the drifting snow sublimation, which occurs in the dry atmosphere. We supposed that model baises in SMB mainly resulted of unresolved drifting snow transport fluxes as they were strongly correlated to the curvature of topography, which drives the wind divergence at the ice sheet surface (Fig. R1) and thus the transport of mass.

[Figure]

**Fig. R1** (a) Curvature of topography computed on the MAR grid (10-6 m-1) (b) Divergence of the mean annual 10 m wind in MAR (m s-1 km-1)

In the revised version of the manuscript we highlight that drifting snow transport might be the first order process when compared to drifting snow sublimation for the four transects we studied, because those 4 transects are located at high elevation places (>2000 m a.s.l.) where the cold atmosphere has low capacity to hold moisture.

We demonstrate this statement by computing the moisture holding capacity in the MAR atmospheric boundary layer (ABL). We re-ran MAR-ERA-Interim for the year 2015 in order to extract variables in the whole atmosphere. With daily variables we compute the moisture holding capacity of the ABL:

Sum_(k=surface↦ABL summit) (Qsat-Q) DeltaP / g

with Q the specific humidity, Qsat the specific humidity at saturation, Delta P the pressure width of the layer and g the gravitational acceleration. We compute the top of the ABL as the level where the turbulent kinetic

energy amounts to 1% of the turbulent kinetic energy maximum in the lowest layers of the model (5% is used in Gallée et al. 2015[1]). We compute Qsat using the relative humidity rh (Qsat = Q/rh).

[Figure]

**Fig R2** (a) Atmospheric boundary layer (ABL) moisture holding capacity in MAR for the year 2015, in kg m-2 yr-1. The ABL moisture holding capacity is computed with daily variables:

ABL moisture holding capacity = Sum_(k=surface↦ABL summit) (Qsat-Q) DeltaP / g

with Q the specific humidity, Qsat the specific humidity at saturation, Delta P the pressure width of the atmospheric layer and g the gravitational acceleration. We compute the top of the ABL as the level where the turbulent kinetic energy amounts to 1% of the turbulent kinetic energy maximum in the lowest layers of the model (5% is used in \citet{Gallee:2015jy}). We compute Qsat using the relative humidity rh (Qsat = Q/rh).

(b) Difference between the ABL moisture holding capacity in MAR and the drifting snow sublimation in RACMO2, for the year 2015, in kg m-2 yr-1 (c) ABL moisture holding capacity in MAR (blue dots) and drifting snow sublimation in RACMO2 (red dots), for the year 2015, in kg m-2 yr-1, as a function of the mean 2 m air temperature in MAR, for the year 2015, in °C. The thin solid blue lines are normalized log-normal distribution of the ABL moisture holding capacity in MAR for 5°C temperature bins around -40°C, -30°C, and -20°C. The thick blue dashed line shows the 95% end of the distributions, and the thick blue solid line is the 99% end of the distributions. The pink line shows a Clausius-Clapeyron-like relationship with temperature: y = exp(-Ls/Rv*(1/ta-1/ta0)+log(subl0)) (in kg m-2 yr-1), with ta the air temperature in K, Ls the enthalpy of sublimation (2.8 10^6 J kg-1), Rv the gas constant of water vapor (461.52J kg-1 K), ta0 = 263.15 K and subl0 = 500 kg m-2 yr-1. (d) Same as (c) but for surface elevation instead of air temperature. Normalized distributions are computed for 500 m bins around 1000 m a.s.l., 2000 m a.s.l., and 3000 m a.s.l.

The ABL moisture holding capacity computed in the MAR model represents the maximum moisture amount that can be loaded in the atmospheric boundary layer according to the MAR simulations. We can confidently consider this ABL moisture holding capacity as an upper bound for drifting snow sublimation amounts (panels a and b), as MAR not including the drifting snow process implies that the ABL keeps its full potential to hold moisture. The ABL moisture holding capacity is exponentially dependent to the air temperature, following a Clausius-Clapeyron-like relationship (panel c).

The ABL moisture holding capacity computed in the MAR model represents the maximum moisture amount that can be loaded in the atmospheric boundary layer according to the MAR simulations. We can confidently consider this maximum moisture content in the ABL a as an upper bound for drifting snow sublimation amounts (R2a -b), as MAR does not include drifting snow processes implying that the ABL keeps its full potential to hold moisture. The ABL moisture holding capacity is exponentially dependent to the air temperature, following a Clausius-Clapeyron-like relationship (R2c).

Fig R3 shows for each of the 4 studied transects the 2 m air temperature, the ABL moisture holding capacity (considered as an estimation of the max. drifting snow sublimation), together with RACMO2 drifting snow

[1] Gallée H., Preunkert S., Argentini S., Frey M.M., Genthon C., Jourdain B., Pietroni I., Casasanta G., Barral H., Vignon E., Amory C., & Legrand M. (2015) Characterization of the boundary layer at Dome C (East Antarctica) during the OPALE summer campaign. *Atmospheric Chemistry and Physics*, **15**, 6225–6236.

sublimation, the drifting snow transport estimate which is function of the curvature, together with RACMO2 drifting snow transport flux, and models SMB biaises. The amplitude of model biases (Delta SMB) is in fact the result of variations of the observed SMB around the smooth simulated SMB gradients (see Fig. 2). These fluctuations, which we called "oscillations", have an amplitude independent of the air temperature, whereas the moisture holding capacity quickly tends to zero when the mean air temperature decreases below -30°C, which is the case along most of the transects. Furthermore, those fluctuation are significantly correlated to the curvature of the topography (Fig 3a). Consequently we are confident that fluctuation of the observed SMB signal for the 4 stake lines is related to the amplitude of the snowdrift transport only.

[Figure]

**Fig R3** For each of the four long transects is shown, from top to bottom, for the year 2015: (top row) 2 m air temperature, in °C; (2nd row) ABL moisture holding capacity in MAR (blue line), and drifting snow sublimation in RACMO2 (red line), in kg m-2 yr-1; (3rd row) the drifting snow transport estimate as a function of curvature (black line), the drifting snow transport simulated by RACMO2 (solid red line), in kg m-2 yr-1; (bottom row) the difference between modeled and observed SMB for MAR (blue line) and RACMO2 (red line), in kg m-2 yr-1. The blue bands are when the curvature of the topography is greater than 0.004 10-6 m-1 (crests) and yellow bands are when the curvature of the topography is lower than -0.004 10-6 m-1 (valleys).

We agree that the drifting snow sublimation might be the largest mass sink in Antarctica, much larger than the drifting snow transport fluxes at the scale of the ice-sheet, but we cannot constrain it with the available observations as drifting snow sublimation occurs bellow 2000 m a.s.l., where observations are extremely scarce. Even if drifting snow transport fluxes are of second order with regards to drifting snow sublimation at the continental scale, our drifting snow transport estimate could be used to constrain the drifting snow mass

transport in models, which might impact the drifting snow sublimation amounts at the ice-sheet margins. E.g., the fact that the drifting snow transport fluxes we estimate are three time larger than those computed by RACMO2 imply in turn than drifting snow sublimation in RACMO2 might be underestimated too.

*Coriolis effect*

With regard to the Coriolis effect, it has indeed an impact on the wind direction. For correlating model SMB biases and curvature we initially introduced a shift of +/- 1 or 2 grid cells along the transects, according to the maximum of correlation between model biases and curvature. In the revised version of the manuscript, we added the deviation of wind flow with regards to the largest topographic slope, which is related to the Coriolis deflection (Fig. R4). We find a magnitude of wind deflection of +/- 1 grid cell, except for Mawson-Lambert Glacier which reaches larger wind deflection. We changed the +2 grid cell shift for Syowa-Dome F for a +1 grid cell shift according to our new computation, and kept a -1 grid cell shift for Mawson-Lambert Glacier. This change for Syowa-Dome F had no impact on the rest of the manuscript.

[Figure]

**Fig. R4** Estimate of the Coriolis deflection of the katabatic wind flow at the ice sheet surface. We compute the angle between the gradient of the topography (direction of the maximum slope) and the wind direction, and convert it in a deflection value, in percentage of the grid box size (deflection = tan(angle)). As transects are shown from the coast to the plateau, the deflection is given a positive sign when the wind is deflected toward the plateau, and a negative sign when the wind is deflected toward the coast along each transect. Consequently, as curvature of the topography is used as a proxy of wind divergence, which drives the drifting snow transport, the shift of the curvature of +/- one grid cell according to the maximum of correlation with SMB bias is in agreement with the Coriolis wind deflection.

**Changes:**

We defined the term "drifting snow transport" in the introduction

We changed the term "erosion-deposition" to the term "drifting snow transport" everywhere.

We removed one sentence which was redundant with the definition

We re-formulated and clarified Section 3.2, renamed "Drifting snow transport features":

- We clarified the link between wind divergence, curvature, and drifting snow transport
- We added the information on the Coriolis effect
- We added figures R1, R2, R3 and R4 in supplementary material.
- We clarified the relative importance of drifting snow transport and drifting snow sublimation above and bellow 2000 m asl.

- Authors have supposed that "snow is usually eroded from topographic crests and collected in the valley", this hypothesis is not corroborated from any field observation (see GPR profile of Talos Dome, Frezzotti et al. 2007, Fujita et al. 2011, Stake profile Syowa-Dome F, Zhongshan-Dome A etc.) in East Antarctica. On the ice divide (crest) the wind are slower and does not eroded/sublimate, whereas in the valley the katabatic wind speed increase and sublimate the drifting/blowing snow. SMB measurements point out that in East Antarctica there are very few evidence of erosion/deposition (less than 10%), most of the process are snow erosion/blowing/sublimation without any redeposition (see Frezzotti et al., 2004, 2007; Scarchilli et al., 2010; Minghu et al., 2011, Ding et al., 2017). Erosion/deposition process can occur on saturated condition at ice shelf or where the slope along wind direction does not permit the sublimation because the atmosphere became saturated soon.

**Answer:**

All our statements are for large scale patterns. The articles you cite have a focus on kilometric-scale ablation-deposition which is driven by the wind speed acceleration/deceleration on kilometric topographic

features. At the larger scales, i.e. ~100 km, the mass transport is also related to the flux divergence, and the intensity of the flux is related to mass available in the atmosphere (thus indirectly to the wind speed).

Thank you very much for sending the GPR accumulation data for Talos Dome of your article Frezzotti et al. (2007). Even if we did not include this transect in our study, because it covers a too long period with regard to the models (~1905-2001 vs. 1979-2010), we show the results here as we think it well illustrates the scale difference in the processes that we are considering (Fig. R5). We find that the Talos transect is located at a high elevation site, with a mean annual temperature between -30°C and -40°C, which means that the atmosphere has a very low capacity of to be loaded with moisture in this sector (Fig. R5c). We also find that the Talos transect is located on a crest (Fig. 5b and c). Consequently we conclude that the mass loss along the transect can be attributed to the drifting snow export process, i.e. the mass export related to the wind divergence at the crest. This is not contradictory with Frezzotti et al. 2007, which find a mass loss along the transect. The analyses of Frezzotti et al. 2007 and Frezzotty et al. 2002, done at the kilometric scale, show that the snow ablation occurs downwind of the kilometer-scale topographic crests. This process is not simulated in the model as those kilometer-scale features are averaged out in the 35 km grid boxes. However it does not exclude that this mass loss is transported further.

**Changes:** We removed the sentence "snow is usually eroded from topographic crests and collected in the valley" and clarified the relationship between curvature, wind divergence, and snowdrift transport in Section 3.2.

[Figure]

**Fig. R5** (top) Analysis of Talos transect ablation patterns in Frezzotti et al. 2007; (bottom left) Map of curvature of the topography computed on the MAR grid, in 10-6 m-1, with location of the Talos transect in red. Dark green colors mark crests, whereas dark brown values mark valleys; (bottom right) as if Fig. R4 but for Talos transect.

- It is not explained why the erosion/drifting module of MAR are not used. Drifting/sublimation snow is very important component of SMB, as also reported by authors.

**Answer:** The drifting snow module in MAR is still under evaluation for its application at the Antarctic scale. We hope our developments will come to end in the forthcoming months. We clarified the sentence in the model description.

**Changes:** We modified the sentence related to the drifting snow module in the model description: "As in \citet{Fettweis:2017de}, the MAR drifting snow scheme is not activated, because this scheme was sensitive to parameter choices \citep{Amory:2015kp}. An updated version of the drifting snow scheme is currently being developed and evaluated for application at the scale of the whole ice sheet.". All references to the MAR drifting snow module were removed elsewhere.

- Authors point out that at regional and continental scale the results of the simulated SMB do not present significant difference and are in good agreement (page 6), despite significant differences components in the negative value of SMB, in absolute value can be correct, but the comparison of the single SMB components are very different.

**Answer:** The main SMB component is the precipitation amount which is of same magnitude in MAR and RACMO2 (2306 ± 111 Gt yr$^{-1}$ in MAR and 2339 ± 107 Gt yr$^{-1}$ in RACMO2 for the ice sheet without peninsula). This is the same for the other basins. The differences in the ablation terms are one to two order of magnitude lower than the SMB and precipitation amounts.

**Changes:** We added snowfall and sublimation amounts for the grounded ice sheet, East Antarctica and West Antarctica.

- Authors must be taking in account the coarse resolution used, in particular in the coastal and confluence area where 35 km of horizontal resolution are too coarse to simulate valley, this influence strongly the wind speed and relative sublimation process.

**Answer:** The relatively coarse resolution is indeed an issue close to mountain ranges, e.g. the Antarctic Peninsula (which we excluded of the analyses) and the Transantarctic mountains. This resolution issue is likely one of the main reason why MAR and RACMO2 diverge in the lee side of the Transantarctic mountains. Far from those specific mountainous areas, the ice sheet topography is rather smooth, and the 35 km seems sufficient to resolve large scale patterns, as demonstrated by the good agreement between modelled and observed SMB patterns. We agree that using higher resolution might improve wind fields at the ice sheet margins and consequently might be of importance for accurately modelling the drifting snow fluxes.

**Changes:** We clarified the role of the resolution in Section 3.4 "Precipitation formation and advection". The role of the resolution is also highlighted in the last sentence of the manuscript.

- Due to the different climatic condition, mainly melt and limited katabatic wind phenomena, the SMB components analysis of the Peninsula, West Antarctica and Ross/Filchner-Ronne ice shelves area should be analysed separately by EAIS.

**Changes:** In Table 2 we included the SMB components for the Grounded ice sheet, Grounded East Antarctica and Grounded West Antarctica. We did not include it for the Peninsula as it is not sufficiently resolved at this resolution in MAR nor in RACMO2. This is why we computed all mass balance also excluding the peninsula.

**Detail:**

- Pag 2 line 6, also MB from GRACE or altimeter use extensively SMB estimation.

**Changes:** Thank you, we added this information : "The total ice sheet mass balance (SMB minus D) can be assessed using satellite altimetry, gravimetry or the input–output method \citep{Shepherd:2018tq}, which all request surface mass balance estimates."

- Pag 3 line 28, "Fresh snow" density cannot be 400 kg m3, use "surface snow"

**Changes:** Thank you, changed.

- Pag. 4 line 12, drifting snow is not a negligible components, and cannot compensate by higher surface sublimation, result from MAR drifting snow should be presented.

**Answer:** We agree that the drifting snow sublimation cannot be compensated by surface snow sublimation in the model, we modified the sentence. Unfortunately the MAR drifting snow module is not ready yet to be applied at the Antarctic scale. The manuscript ends on the conclusion that including drifting snow in MAR is of importance.

**Change:** We changed the sentence explaining why the MAR drifting snow routine was not activated.

- Pag 6 line 24 76 kg/m2/yr is not a negligible value and represent about 60%!!! Please comment and integrating.

**Answer:** You're right, we did a mistake by comparing the RMSE to the mean value whereas it should rather be compared with the standard deviation. We found it is easier to interpret the correlation coefficient, so we replaced the RMSE by the correlation coefficient of the log(SMB) (SMB distribution are log-normal)

**Change:** We replaced the RMSE by the R2 of log(SMB)

- Pag 6-7-8-9-10 see above main comments

See answers and changes above.

- Pag 9 table 2 The different component of SMB must be tabled in different way, positive component: snowfall and rainfall; negative term: sublimation and run off; surface process: melt-refreezed into the snowpack and erosion-deposition.

**Change:**

We changed the table according to your suggestion, thank you.

We specified how SMB is computed in the table legend.

We corrected discrepancies between the SMB and its individual components for RACMO2, which were caused by the interpolation method.

We added snowfall and sublimation fluxes for the major basins.

- Pag 10 line 8-12 erosion-deposition is a "sedimentation" phenomenon, if snow sublimate and then redeposit under snowfall it is not exported in atmosphere/ocean, rewriting the text.

**Answer:** Following your comments, we thought that the term "erosion-deposition" was misleading as erosion is usually interpreted as the removal of snow from the snowpack to the atmosphere, whereas the flux we wanted to describe was the horizontal advection of drifting snow. As katabatic winds flow toward the ocean, a part of the drifting snow mass is advected through the ice sheet boundary and is consequently exported outside the ice sheet.

**Changes:**

We defined the term "drifting snow transport" in the introduction.

We changed the term "erosion-deposition" to the term "drifting snow transport" everywhere.

We removed one sentence which was redundant with the definition.

- Pag 11 line 5 I do not understand, MAR drifting module is used or not, why several repetition about MAR drifting module?

**Answer:** We did not use the drifting snow module.

**Changes:** We removed all references to the drifting snow module in MAR except in the model description Section 2.1.1. so that it is clearer that we did not used this module.

- Pag 12-13 The Grazioli paper is very interesting, but snowfall generally occurs under cyclonic storm and not under "pure" katabatic wind phenomena. Katabatic wind arrives later with strong blowing snow phenomena and related sublimation (see Palm et al., 2011, 2017; Scarchilli et al., 2010). Wind during cyclonic storm are variables and not from dry high-elevated inland plateau toward sea level. This does not exclude that wind sublimation occur during a storm, but normally during marine storm the atmosphere is already saturated with low capacity of sublimation.

**Answer:** We re-ran the MAR model for the year 2015 (same year as in Grazioli et. al. 2017), and saved the 3D snowfall component, as requested by Referee#2. This allowed us to compute the sublimation of the precipitation in the katabatic layer in MAR. We find a similar atmospheric sublimation amount as in Grazioli et al. 2017, which cannot be associated with the drifting snow sublimation, as the drifting snow process is not included in MAR simulation. Even if not every cyclonic storm bring humidity toward the ice sheet above the katabatic layer, from an observational basis (Grazioli et. al. 2017) and from our modelling study, this phenomena seems important enough to significantly impact the precipitation amount at the surface.

**Changes:** New map of atmospheric sublimation in MAR and RACMO2 (Fig.5c) and new estimates of the atmospheric sublimation modelled by MAR compared to RACMO2 and Grazioli et al (2017) in Section 3.3.

- Pag 14 line 3-7 Wind crust area reported in Scambos et al. 2012 are related to hiatus in accumulation driven by sublimation wind process, it is not clear the relation with observed difference between MAR and RACMO2 snowfall. The wind crust is the extreme phenomena where the ratio between snowfall and wind sublimation conduct to hiatus in accumulation from several to thousand year (see Frezzotti et al., 2002, 2005). Due to the limit of method of Scambos et al., 2012, wind crust are surveyed only in the inland plateau (above 1500 m) where the coarse resolution of models have less impact on the slope along wind direction and therefore wind speed. Wind crust is the upper limit of hiatus, before became blue ice area, but they represent only a limited area of wind drive sublimation area (see Palm et al. 2011, Frezzotti et al., 2007; Minghu et al., 2011) those are more extended than permanent wind crust surface mapped by Scambos et al., 2011. Models firstly must be reproduce the wind crust hiatus, if they can be representative of the negative term of SMB.

**Answer:** Thank you for your analysis and the references. After a more detailed reading of the literature, and as you state, it appears that the relationship between wind glaze areas and drifting snow sublimation is not

straightforward. As wind glaze are concomitant with megadunes at a kilometric scale, they might not be systematically associated with a mass loss at the scale of a model grid box. In particular, we identified that a large portion of the mapped wind glazes (Scambos et al., 2012, nicely shared by Ted Scambos) was located in areas of very low temperature, where the atmosphere has very low potential to be loaded with moisture. Consequently we removed our analysis on wind glaze areas, and put more emphasis of the potential mass loss by drifting snow sublimation at the ice sheet margins.

**Changes:** We removed the reference to wind glaze areas in Section 3.4 and in the discussion. We put more emphasis of the potential mass loss by drifting snow sublimation at the ice sheet margins in Section 3.3.

- If it could be useful, the SMB Talos Dome transect published on Frezzotti et al., 2007 paper is available for the comparison of models.

**Answer:** Thank you for sending the Talos Dome GPR data. See the analysis on Talos Dome above.

**Reference:**

Das, I., Bell, R. E., Scambos, T. A., Wolovick, M., Creyts, T. T., Studinger, M., ... & Van Den Broeke, M. R. (2013). Influence of persistent wind scour on the surface mass balance of Antarctica. Nature Geoscience, 6(5), 367.

Das, I., Scambos, T. A., Koenig, L. S., Broeke, M. R., & Lenaerts, J. (2015). Extreme wind  Rice interac- ˘ tion over Recovery Ice Stream, East Antarctica. Geophysical Research Letters, 42(19), 8064-8071.

Ding, M., Zhang, T., Xiao, C., Li, C., Jin, B., Bian, L., ... & Qin, D. (2017). Snowdrift effect on snow deposition: Insights from a comparison of a snow pit profile and meteorological observations in east Antarctica. Science China Earth Sciences, 60(4), 672-685.

Fujita, S., Holmlund, P., Brown, I., Enomoto, H., Fujii, Y., Fujita, K., ... & Hoshina, Y. (2011). Spatial and temporal variability of snow accumulation rate on the East Antarctic ice divide between Dome Fuji and EPICA DML. The Cryosphere, 5(4), 1057.

Frezzotti, Massimo, et al. "Snow dunes and glazed surfaces in Antarctica: new field and remote-sensing data." Annals of Glaciology 34 (2002): 81-88.

Frezzotti, M., Pourchet, M., Flora, O., Gandolfi, S., Gay, M., Urbini, S., ... & Severi, M. (2004). New estimations of precipitation and surface sublimation in East Antarctica from snow accumulation measurements. Climate Dynamics, 23(7-8), 803-813.

Frezzotti, M., Pourchet, M., Flora, O., Gandolfi, S., Gay, M., Urbini, S., ... & Severi, M. (2005). Spatial and temporal variability of snow accumulation in East Antarctica from traverse data. Journal of Glaciology, 51(172), 113-124.

Frezzotti, M., Urbini, S., Proposito, M., Scarchilli, C., & Gandolfi, S. (2007). Spatial and temporal variability of surface mass balance near Talos Dome, East Antarctica. Journal of Geophysical Research: Earth Surface, 112(F2).

Minghu, D., Cunde, X., Yuansheng, L., Jiawen, R., Shugui, H., Bo, J., & C4 TCD Interactive comment Printer-friendly version Discussion paper Bo, S. (2011). Spatial variability of surface mass balance along a traverse route from Zhongshan station to Dome A, Antarctica. Journal of Glaciology, 57(204), 658-666.

Scambos, T. A., Frezzotti, M., Haran, T., Bohlander, J., Lenaerts, J. T. M., Van Den Broeke, M. R., ... & Neumann, T. (2012). Extent of low-accumulation'wind glaze'areas on the East Antarctic plateau: implications for continental ice mass balance. Journal of glaciology, 58(210), 633-647.

Palm, S. P., Yang, Y., Spinhirne, J. D., & Marshak, A. (2011). Satellite remote sensing of blowing snow properties over Antarctica. Journal of Geophysical Research: Atmospheres, 116(D16).

Palm, S. P., Kayetha, V., Yang, Y., & Pauly, R. (2017). Blowing snow sublimation and transport over Antarctica from 11 years of CALIPSO observations. The Cryosphere, 11(6), 2555. Scarchilli, C., Frezzotti, M., Grigioni, P., De Silvestri, L., Agnoletto, L., & Dolci, S. (2010). Extraordinary blowing snow transport events in East Antarctica. Climate Dynamics, 34(7-8), 1195-1206.

——————————————————————————————————-
**Anonymous Referee #2**

——————————————————————————————————-

**General comments:**

This paper presents performance of the polar regional climate model MAR applied in the entire Antarctic Ice Sheet (AIS) for the first time. MAR has been applied and validated in the Greenland ice sheet (GrIS) for a long time, and it is widely recognized as a useful and reliable tool to understand the polar climate system. In the present study, the authors follow basically the same MAR model configuration developed in the GrIS. In addition, they decrease horizontal and vertical resolutions (due to the AIS's much larger area than the GrIS), use a boundary relaxation of upper air temperature and wind speed, employ an optimized fresh snow density parameterization for the AIS, and utilize a dynamic parameterization for the aerodynamic roughness length.

This reviewer finds that these modifications are reasonable to conduct this kind of study.

The model forced by the European Centre for Medium-Range Weather Forecasts (ECMWF) Interim reanalysis (ERA-Interim) is evaluated in terms of surface mass balance (SMB) using the in-situ data obtained during 1979-2015. In this process, the authors also refer to model simulation results by another polar regional climate model known as RACMO2 (horizontal resolution is 27 km) forced by the same reanalysis data to identify important physical processes that influences the AIS SMB simulations. The authors find that both models tend to accumulate too much snow on crests, whereas not enough snow in valleys. Here, the authors attribute the main reason for this discrepancy to the insufficiency of drifting snow-induced erosion-deposition process modeling in both models. When calculated SMBs by MAR and RACMO2 are integrated over the AIS, no significant differences are found between these two results; however, geographical SMB patterns for both models differ significantly, which suggest that there are many things to do to develop a truly reliable polar regional climate model for the AIS. In valleys, RACMO2-simulated precipitation is larger than that by MAR: it is mainly attributed to a difference in modeling approach for sublimation in unsaturated katabatic layer. On the other hand, larger precipitation in the inland AIS is simulated by MAR, because of the difference in horizontal resolution set in both models, which significantly affect orographic impacts on the simulated precipitation rates.

Overall, this paper is well written and can be informative for readers who are interested in the AIS climate system; however, this reviewer thinks that some discussions are not deepened sufficiently and suggests the following points to be considered before the publication. In the following part, this reviewer gives specific comments. Page and line numbers are denoted by "P" and "L", respectively.

Dear Referee, we thank you for your detailed and nice summary and for your useful comments bellow.

——————————————————————————————————

**Specific comments (major)**

P. 9, L. 3: What do the authors mean by "shift" mentioned here? Why is this procedure needed here? Please explain more about the procedure.

**Answer:** Following your comment and the one of Referee#1 (M. Frezzotti) about the consideration of the Coriolis effect, we introduced a more in-depth justification for the "shift", and a better description of the procedure in Section 3.2.

**Changes:**

We changed the sentence for the following: "To quantify this curvature effect, we correlate MAR SMB bias (\Delta SMB) with the curvature. For each transect, we apply a constant shift of +/- one grid cell to the curvature in order to find the maximum correlation with \Delta SMB (Fig.~S9). The sign and the amplitude of these shifts are in line with curvature being used as a proxy for wind divergence, as they are consistent with the Coriolis wind deflection westward of the topography gradient (detailed in Fig.~S10)."

We added the Fig.S10 detail the Coriolis deflection and its relation with the shift of curvature.

P. 11, L. 1.: The authors mention that near surface atmosphere is simulated to be drier in MAR compared to RACMO2. How large is the difference? Please quantify and discuss why the difference was made.

**Answer:** This statement is based on a physical consideration: given that RACMO2 includes drifting snow whereas MAR does not, it implies that the sublimation of drifting snow particles are allowed to sublimate and thus to moisten the surface atmospheric layer in RACMO2, consequently reducing the sublimation at the ice sheet surface. Unfortunately we cannot quantify this phenomenon as the larger surface sublimation in MAR might have reduced the differences between MAR and RACMO2 moisture content in the surface atmospheric layers.

**Changes:** We better specified in the text that this statement was based on a physical consideration and not on a quantified basis in Section 3.2: "Drifting snow sublimation included in RACMO2 and not in MAR moisten

the surface atmospheric layers, consequently reducing the sublimation at the surface of the snowpack. This might explains the stronger surface snow sublimation in MAR than in RACMO2 (Table~\ref{tab:2} and Fig.~S5)."

Sect. 3.3: Using MAR, can the authors perform a model sensitivity test where sublimation in unsaturated katabatic layer is not allowed? If results from this sensitivity test are provided, the argument by the authors in this section would become more convincing.

**Answer:**

As requested we performed new simulations for quantifying the amount of sublimation in the katabatic layer in MAR. We did not prevent the saturation to occur in the atmosphere because it could induce feedbacks with the dynamics and the surface energy balance, but we extracted the 3D snowfall fields in the atmosphere and computed the low-level sublimation as in Grazioli et al. 2017:

for daily outputs, atmospheric sublimation = (maximum precipitation in the atmosphere) - (precipitation at ground).

We found that MAR sublimates precipitation in the katabatic layers with similar amounts to those found in Grazioli et al. 2017.

Meanwhile, we discovered that RACMO2 did sublimate the precipitation in the atmosphere, contrary to what the authors initially stated. We added one co-author in the RACMO team, W.J. van de Berg, who computed the amount of atmospheric sublimation in RACMO2.

We conducted new analysis based on the maximum precipitation amounts in MAR and RACMO2 and concluded about differences in precipitation patterns being due to differences in the precipitation advection inland.

**Changes:**

Section 3.3 was updated according to our new quantification of the atmospheric sublimation in MAR and RACMO2 for the year 2015.

Section 3.4 was modified to focus on the difference in SMB patterns between MAR and RACMO2, which we attribute to precipitation formation timing and advection.

Sects. 3.3 and 3.4: In Sect. 3.4, the authors point out the importance of orographic effects on the precipitation simulations in areas centered on crests. It is interesting the authors don't mention orographic effects on the precipitation simulations at valleys (Sect. 3.3). Do the authors think that considering the process for the low-level sublimation in unsaturated atmosphere (at especially valleys) is more important than setting a higher horizontal resolution to obtain realistic SMB at valleys by a model?

**Answer:** You're right. The role of the topography resolution might induce a difference in the precipitation advected inland, which could contribute to the difference in maximum snowfall amounts between MAR and RACMO2.

**Changes:** We added this analysis at the end of Section 3.4.

P. 13, L. 33: Can the authors perform a MAR model sensitivity test where the horizontal resolution is set to be 27 km (same as RACMO2) or higher? I know it is computationally demanding, but, results from such a sensitivity test for even only several years would be informative for readers.

**Answer:** We agree that including sensitivity tests on resolution would have been an added value for this study, but unfortunately we did not have time to do them. Given the potential importance of the atmospheric sublimation, we preferred to re-run MAR for 2015 to better quantify this process, as you requested above. However we are planning to run MAR at different resolutions in a future study, which is highlighted in the last sentence of the manuscript.

————————————————————————————————————

**Specific comments (minor)**

P. 2, L. 4: Regarding the "several approaches", please list up and explain these approaches briefly here. I believe the information are very informative for readers.

**Answer:** We agree.

**Changes:** We added the information: " The total ice sheet mass balance (SMB minus D) can be assessed using satellite altimetry, gravimetry or the input–output method \citep{Shepherd:2018tq}, which all request surface mass balance estimates. The input-output method, which consists in separately modelling ice dynamics and surface mass balance, is also the only way to project future trends."

P. 3, L. 18: Why did the authors set the horizontal resolution to be 35 km for MAR in the present study? To perform detailed and solid comparisons between MAR and RACMO2, setting the same horizontal resolution is very ideal.

**Answer:** The primary reason is that we did not set-up the model with the aim to compare our simulations to RACMO2, but rather to find a good compromise for being able to run MAR with multiple forcings (here: 3 reanalyses, and in the future: GCMs from CMIP5 and CMIP5) over decadal to centennial time scales. A 27 km resolution would have requested a too large increase in computational time. In addition, it is not only the resolution which is different between MAR and RACMO2, but also the grid projection, which cannot be changed neither in MAR nor in RACMO2.

P. 5, L. 10: Figure 1 basically presents simulation results from MAR, therefore, referring Fig. 1 in this sentence is a bit strange (MAR simulation results don't reproduce the reality, although I agree it certainly does a good job.).

**Changes:** We added the observed SMB in Fig.1a.

P. 6, L. 8 ~ 10: I could not follow the explanation here. Could you please detail more?

**Answer:** We keep observations beginning before 1979 only if they cover more than eight years, and in this case we compare the observed value with the modelled value time-averaged for 1979-2015.

**Changes:** We changed the sentence.

P. 6, L. 23: For me, it is not easy to understand the authors' intension regarding "oscillates" mentioned here. Could you please reformulate it?

**Answer:** We agree it was not clear, we wanted to say that MAR SMB shows no systematic spatial bias.

**Changes:** We changed the sentence

P. 6, L. 23 ~ 24: In Sect. 3, the authors present the performance of modeled SMB by MAR. They also perform detailed comparisons between simulation results from MAR and RACMO2. In this context, I think it is better to denote the performance of RACMO2 in terms of SMB here in the same manner as MAR (please indicate mean bias and RMSE for RACMO2).

**Changes:** We added the information.

P. 7, L. 10: It is not easy to understand the meaning of "oscillations" mentioned here. Could you please rephrase it?

**Changes:** We changed "oscillations" for the more accurate word "fluctuations".

P. 13, L. 4 ~ 14: Do the authors mean that the MAR-simulated precipitation at valleys is more realistic compared to the RACMO2-simulated precipitation at valleys? Please describe more clearly.

**Answer:** yes

**Changes:** We explicitly stated that RACMO2 likely underestimate the atmospheric sublimation: " A major difference between MAR and RACMO2 is the advection of precipitation in the atmosphere: in MAR, precipitating particles are explicitly advected through the atmospheric layers until they reach the surface, while in RACMO2, precipitation is added to the surface without horizontal advection, and is able to interact with the atmosphere in a single time step only (6 min in this simulation). Consequently, atmospheric sublimation is likely to be underestimated in RACMO2."

P. 14, L. 3: "wind glaze area": Please detail more about its definition here.

**Answer:** We decided to remove all reference to wind glaze. We re-write here an answer made to Referee#1: After a more detailed reading of the literature, it appears that the relationship between wind glaze areas and drifting snow sublimation is not straightforward. As wind glazes are concomitant with megadunes at a kilometric scale, they might not be systematically associated with a mass loss at the scale of a model grid box. In particular, we identified that a large portion of the mapped wind glazes (Scambos et al., 2012, nicely shared by Ted Scambos) was located in areas of very low temperature, where the atmosphere has very low potential to be loaded with moisture.

**Changes:** We removed all reference to wind glaze.

————————————————————————————————

Technical corrections:
Figure 1: Please explain red circles in Figs. 1a to 1c in the caption. It is also the case for Figs. 4b and 4c.

**Changes:** corrected.

P. 9, L. 5: "wind speed" -> "10 m wind speed"?

**Changes:** thank you, corrected.

P. 13, L. 22 ~ 23: In Fig 5b, no description on the altitude of the AIS is provided. Please check it again and revise it.

**Answer:** All this section was deeply revised following the new simulation requested. This sentence was removed and their is no reference to the curvature in Fig5 anymore.

---

## Author Comment (AC4) · 29 Oct 2018

**Estimation of the Antarctic surface mass balance using MAR (1979-2015) and identification of dominant processes**

Cécile Agosta1,2,3, Charles Amory1, Christoph Kittel1, Anais Orsi2, Vincent Favier3, Hubert Gallée3, Michiel R. van den Broeke4, Jan T. M. Lenaerts4,5, Jan Melchior van Wessem4, Willem Jan van de Berg4, and Xavier Fettweis1

[revised manuscript text omitted]

---

## Author Response (AR2)

**Dear Kenny Matsuoka, TC editorial board, and reviewers,**

**We thank you a lot for your useful comments and suggestions. You can find bellow our modifications to the manuscript following your recommandation (in bold). We hope we answered all your comments, but we would be happy to clarify further the manuscript if needed.**

**With our best wishes for the upcoming new year,**

**Cécile Agosta, on behalf of all co-authors**

The revised manuscript is associated with 13 supplemental figures and 2 tables. This helps readers to get the comprehensive understanding of this work, but it requires a lot of cross-referring. In some cases, extra few sentences in the main text could help readers understand the contents without cross-referring the supplement. Please review the entire manuscript to see (1) whether arguments in the supplement are cited at the right position (rather than as an addition at the end of the discussion for completeness), and (2) whether the main text summarizes the materials presented in the supplement (if not, add a few sentences to help readers). This is the case for snow drift analysis using surface curvature and wind speed. When I read the manuscript pages 11-12 and see Figures 4, S11 and S12, I was not convinced why these wind threshold values are chosen. Then, at P12 L19, Table S2 is refereed, in which I found more detail analysis for different wind speed thresholds.

**We clarified the paragraph about wind thresholds :**

**"We propose that drifting snow transport fluxes ($ds_{tr}$) not resolved by MAR can be estimated as a scaling of curvature depending of wind speed: $ds_{tr} = a(ws_{10}) \cdot$ curvature (Figure 4b). The scaling factor $a(ws_{10})$ depends on wind thresholds to simulate the transition between no drifting snow transport for low wind speed ($a = 0$ for $ws_{10} < 5$ m s$^{-1}$) and drifting snow transport scaled to curvature for high wind speed ($a = 3700\ 10^6$ kg m$^{-1}$ yr$^{-1}$ for $ws_{10} > 9$ m s$^{-1}$), with a linearly increasing scaling factor between 5 and 9 m s$^{-1}$ for a smooth transition around the 7 m s$^{-1}$ wind threshold defined above. That estimate of drifting snow transport fluxes shows little sensitivity to the choice of the wind thresholds and of the scaling factor (see fluxes summed over the ice sheet for different thresholds and scaling factors in Table S2). The spatial pattern of drifting snow transport we obtain is comparable to the one simulated by RACMO2 (Fig. 4c), except that it gives fluxes more than three times larger than in RACMO2 (see Table S2, and note the different colour map scales between Fig.4b and 4c)."**

**We better summarized the material presented in Fig. S4 and S5 (ex. Fig. S13), Fig. S9, and Fig. S10:**

**"Another noticeable result is that MAR forced by ERA-Interim, JRA-55 and MERRA2 give very similar results for the SMB spatial pattern, not only at the observation locations (Fig. 2) but also at the ice sheet scale (comparisons of MAR SMB for different forcing reanalyses are shown in Fig.S4, with colormap scales 10 time smaller than in Fig.S5 where MAR is compared to RACMO2)"**

**"As a consequence, the surface wind divergence, which drives the snowdrift mass transport, is strongly related to the curvature of the topography, and both have similar spatial patterns (shown in Fig.S9)"**

**"For three out of the four transects we find only one shift for which the correlation is significant, and for remaining transect (Syowa--Dome F) we find no significant correlation (Fig.S10)."**

I found many typos; I actually spent a lot of time to realize some of them can be a typo. One reviewer also pointed out that the reference list has typos and should be carefully checked by the authors. Because of the degree of cross-referring in this manuscript, readers can be easily confused with typos.

**[We corrected numerous typo in the text thanks to your suggestions bellow, and in the reference list which has been carefully checked]**

I request minor revision; it means that the new manuscript will not be sent to the reviewers but assessed by the editor. Please submit marked manuscript to highlight changes made in the next round. Point-to-point responses are necessary if you don't agree the suggestions or if you change the manuscript largely different from the suggestions.

The manuscript improved significantly with helpful reviews. Thus, I'd suggest the authors to acknowledge these reviewers.

**[We agree, thank you for your suggestion.]**

Thanks for submitting your work to TCD and I look forward seeing the revised manuscript.

Kenny Matsuoka, TC/TCD Editor

Editorial points:

- P4L1: add unit for snow density. **[corrected]**

- P5L15: what does the hyphen between Fig. 1 and 5% mean? Should it be replaced with comma? **[yes, corrected]**

- P6 Table 1 caption and forward: elevation reference is not shown. Is it reference to the sea level? I am asking this question because the elevations are shown very precisely (to 1 m). If it is referenced to the sea level, rewrite "m" to "m a.s.l." This comment is applicable for all of this problem found elsewhere in the manuscript. **[yes, corrected]**

- P7L5: "scales compared to Fig. S9". Is it Fig. S8, not S9? **[corrected, it was S13 … + more detailed explanation]**

- P8 Figure 2 caption 2-3 lines from the bottom: I cannot see markers with white and black faces. **[clarified]**

- P9 Table 2: Please show the boundary of West and East Antarctic Ice Sheets, as well as the southern boundary of Peninsula in one of maps, probably Fig. 2.

**Basins are now shown in Fig.2 and the information is added in Table 2 caption and in Fig. 2 caption.**

- P10 Fig. 3: "difference in SMB between models and observation" is ambiguous. Please clarify whether it is "(model) – (observation)" or "(observation) – (model)". **[clarified]**

- P10 Fig. 3: Why does MAR SMB is shifted by -30? Does it mean "0" in the figure means "-30"? If this adjustment is made for the presentation purpose, please do not shift the data but change the y axis label. The green and brown masks are used to show positive and negative values. However, because of this shift, the mask may give a wrong impression. My recommendation is to remove this shift, and expand the y axis range to capture the full variations.

**We removed the shift.**

- P10L2: Add "local" before "fluctuations" at the beginning of the paragraph. **[corrected]**

- P11 Fig 4 caption: Do you mean "back outlines" by saying "black contours"? **[corrected]**

- P11 Fig. 4 caption line 7th from the top: it is said "wind speed greater than 9 m/s". Is this typo, and supposed to be "greater than 7 m/s"? (I actually spent long time to realize that it can be a typo and wondered why you use two different criteria).

**There is no typo but we wanted a smooth increase of the alpha coefficient from 0 (wind speed too weak to induce drifting snow) to 3700 (high wind speeds). As we saw that 7 m/s might be a good threshold for wind speed, we computed this smooth increase around 7 m/s. This explains the 5 to 9 wind speed range, and also why we also use the 6-8 wind speed range in Table S2. We added "and α linearly increasing as a function of wind speed in between*, around the 7 m s$^{-1}$ wind speed threshold.*". We also clarified this point in the main text (see bellow).**

- P12 L3-4: here it is said that there is no relationship between SMB and curvature if wind speed is lower than 7 m/s. However, in Fig 4 caption, the drift snow is proportional to the wind speed when the wind speed ranges between 5-7 m/s. Please clarify.

**It is for the same reason than above, I hope the change clarify this.**

- P12 L6: not elevation curvature, but surface curvature. **[corrected]**

- P12 L19-24: unclear. Revise.

**We re-wrote the paragraph, we hope it's now clearer:**

*"Our drifting snow transport estimate gives a good constraint for drifting snow fluxes above 2000 m a.s.l., where low temperatures induce negligible atmospheric sublimation. As drifting snow transport is proportional to the amount of snow in suspension in the atmosphere, quantifying this flux also enables to constrain the amount of snow eroded from the snowpack to the atmosphere, which drives drifting snow sublimation fluxes at lower elevation. This is of importance as drifting snow sublimation is a much larger mass sink than drifting snow transport over the whole ice sheet (Palm et al., 2017; Lenaerts et al.,2012a) but is still poorly constrained because observations are very scarce bellow 2000 m a.s.l. where it occurs."*

- P12 L31: I am not aware that "dynamical downscaling of ERA-interim with RACO2 and MAR" is described above (or what do you want to say with "dynamical downscaling"?). Revise. And cite key figures supporting this sentence. **[corrected]**

**"dynamical downscaling" is another term for "regional climate modelling", I replaced it by the second one, more usually used .**

- P13 Fig. 5c: why do you define the difference as RACMO2 – MAR for this panel, though all the other three defines the difference as MAR-RACMO2. If you don't have a good reason, please use the same definition MAR-RACMO2 for all panels.

**I changed for MAR-RACMO2 in panel c). The reason was that sublimation is a negative contribution to SMB and precipitation, so mapping RACMO2 - MAR for sublimation allows to directly compare the contribution of sublimation to SMB and ground precipitation shown in a) and b). But finally I agree it is clearer to plot MAR-RACMO2 everywhere.**

- P13L2: In this study -> "In that study" or "In Grazioi et al (2017)" **[corrected]**

- P14L14: change to "2000 m a.s.l." **[corrected]**

- P14L27: change to "Figs. 5b and 5d" **[corrected]**

- P15 Fig. 6 y axis labels: unit for panels b and c are wrong. m-2, not m2.**[corrected]**

- P15 Fig.6 caption: line 2 from the top, change to "Figs. 5b and 5d". **[corrected]**

- I think both models use the same BEDAMP2 ice topography. So, why is the surface elevation different (apparent for 1700 km from B1)?

**RACMO2 uses Bamber (2009) whereas MAR uses Bedmap2 (Fretwell et al., 2013). We added this information in the model description, and we also discuss it in the conclusion as you suggest bellow.**

- Add unit for the curvature (0.005 and -0.005). **[corrected]**

- P16L5: again what does "dynamical downscaling" mean? Do you mean individual components of the climate models enforced by these reanalysis data? **[corrected]**

- P16L17: it is first time for me to see that the RACMO2 underestimates the drifting snow transport by a factor of three. It was not quantified earlier, and the factor of three suddenly appears in the conclusions and then abstract. In my opinion, Figure 4 cited here does not immediately support this statement. Please explain.

**This statement is based on Section 3.2 (P12 L14-17 of first revised manuscript) :**

*"In Figure 4b, we propose a spatial estimate of the drifting snow transport fluxes not resolved by MAR, computed as a simple function of curvature and wind speed as described above. This estimate is comparable to the drifting snow transport pattern modelled by RACMO2 (Fig. 4c), except that it gives fluxes approximately three times larger than in RACMO2 (see differences in colour map scales between Fig. 4b and 4c, fluxes summed over the ice sheet and associated uncertainties are detailed in Table S2). "*

**This section is now more detailed :**

*"We propose that drifting snow transport fluxes ($ds_{tr}$) not resolved by MAR can be estimated as a scaling of curvature depending of wind speed: $ds_{tr} = a(ws_{10}) \cdot$ curvature (Figure 4b). The scaling factor $a(ws_{10})$ depends on wind thresholds to simulate the transition between no drifting snow transport for low wind speed ($a = 0$ for $ws_{10} < 5$ m s$^{-1}$) and drifting snow transport scaled to curvature for high wind speed ($a = 3700 \ 10^6$ kg m$^{-1}$ yr$^{-1}$ for $ws_{10} > 9$ m s$^{-1}$), with a linearly increasing scaling factor between 5 and 9 m s$^{-1}$ for a smooth transition around the 7 m s$^{-1}$ wind threshold defined above. That estimate of drifting snow transport fluxes shows little sensitivity to the choice of the wind thresholds and of the scaling factor (see fluxes summed over the ice sheet for different thresholds and scaling factors in Table S2). The spatial pattern of drifting snow transport we obtain is comparable to the one simulated by RACMO2 (Fig. 4c), except that it gives fluxes more than three times larger than in RACMO2 (see Table S2, and note the different colour map scales between Fig. 4b and 4c)."*

- P16L24-25: Revise.

**We revised the sentence for the following:**

**"We also point out that MAR generally simulates larger SMB and snowfall amounts than RACMO2 inland, particularly on the lee side of the Transantarctic Mountains and on crests at the ice sheet margins, whereas MAR simulates lower snowfall than RACMO2 windward of mountain ranges and promontories."**

- P16-17: I agree with the authors on the recommendations of future modeling work. Sublimation and thus wind speed should be better modeled for low elevated regions near the coast, where the surface topography is highly variable and BEDMAP2 topography is not accurate enough. So, model cell size and input topography data should be considered as well. Just a comment.

**We added the following sentence at the end:** *"The accuracy of the topography has to be considered as well, as digital elevation models are in constant improvement over the Antarctic ice sheet (e.g. Slater et al., 2018) and should be regularly updated in climate models."*

Editorial points in the supplement

**We moved Fig.S13 to Fig.S5 according to its first citation in the manuscript, and we changed all the following numbering throughout the manuscript and supplement.**

- Table S1 caption: unclear. Please revise what each number means. I think that the second number shows the number of model cells where the data are present for the specific depth range, and the first number shows the number of observations in total. For example, for 0-100 cm of Albert et al. (2007), only 1 cell has the observation data, and this cell has three data points.

**Yes you are right. We changed the caption for the following: "*References of snow density datasets. For each depth range, we give the total number of observations (left) and the number of 35x35 km model grid cells they cover (right)."***

- Figure S1 caption: Table S1, not S2. **[corrected]**

- Figures S5 and S6 **[now S6 and S7]**: revise the unit to kg m-2 yr-1 in each panel. **[corrected, and also for S11...]**

- Figure S6 **[now S7]** caption: Table 2, not 1. **[corrected]**

- Figure S9 **[now S10]** caption: contour lines -> outlines. **[corrected]**

- Add a brief explanation why these data were excluded. Are they excluded because of the slow wind speed (< 7 m/s)?

**The 2 excluded dots where excluded because they where outliers, and the 6 squares because of the low wind speed at those locations. I clarified this in the legend.**

- Again, why do you use this 7-m/s criteria, though the regression is made for the locations where wind speed is more than 9 m/s? (it may be a typo, however).

**This is the same explanation as above : we chose the 5/9 m s-1 thresholds to have a smooth transition around the 7 m s-1 threshold.**

- Figure S10 **[now S11]**: the definition of positive and negative wind deflection is not clear. Is it better to define as eastward or westward?

**It is defined this way because in Fig.3 we plot the variables against the distance along transect, from the coast (left) to the plateau (right). So the Coriolis deflection must be counted along this same axis : a deflection toward the coast shifts the wind backward in the axis (negative deflection), and a deflection toward the plateau shifts the wind upward in the axis (positive deflection). This information has been added to the legend.**

- Figure S11 **[now S12]**: unit for the gas constant of water vapor should be J kg-1 K-1. Add "-1" after K. **[corrected]** Also thin normalized curves in Panels c and d are hardly readable. **[corrected]**

- Figure S11 **[now S12]**: Please add explanations for 95% and 99% envelopes.

**In the caption it is said: *"The thick blue dashed line shows the 95% end of the distributions, and the thick blue solid line is the 99% end of the distributions."***

- Figure S12 **[now S13]**: check the very end of the caption. The curvature for the valleys must be wrong. **[corrected]**

[revised manuscript text omitted]

December 27, 2018

Table S1: References of snow density datasets. For each depth range, we give the total number of observations (left) and the number of 35×35 km model grid cells they cover (right).

| Reference | Dataset | 0–20 cm | 0–50 cm | 0–100 cm |
|---|---|---|---|---|
| Albert et al. (2007) | SUMup17 [1] | 3/1 | 3/1 | 3/1 |
| Brucker and Koenig (2011) | SUMup17 [1] | 6/5 | 6/5 | 6/5 |
| Cameron et al. (1968) | Kaspers04 [2] | 0/0 | 0/0 | 22/22 |
| Ding et al. (2011) | CHINARE | 568/39 | 0/0 | 0/0 |
| Fujiwara and Endo (1971) | JARE69 | 65/38 | 0/0 | 13/13 |
| Gallet et al. (2011) | DC-DDU08 | 8/8 | 7/7 | 0/0 |
| Herron and Langway (1980) | Kaspers04 [2] | 0/0 | 1/1 | 1/1 |
| Kaspers et al. (2004) | Kaspers04 [2] | 0/0 | 2/2 | 2/2 |
| Kreutz et al. (2011) | SUMup17 [1] | 1/1 | 1/1 | 1/1 |
| Medley et al. (2013) | SUMup17 [1] | 1/1 | 3/3 | 2/2 |
| Sugiyama et al. (2012) | JASE07 | 0/0 | 43/43 | 43/42 |
| Watanabe (1975) | JARE70 | 6/1 | 6/5 | 8/5 |
| van den Broeke et al. (1999) | Kaspers04 [2] | 0/0 | 8/8 | 8/8 |

[1] Montgomery et al. (2018), [2] Kaspers et al. (2004)

Table S2: Estimates of drifting snow transport fluxes summed over the total (TIS, $13.4\ 10^6$ km$^2$) and the grounded (GIS, $12.0\ 10^6$ km$^2$) Antarctic ice sheet, excluding Peninsula. Parenthesis ($\alpha_{max},ws_{min},ws_{max}$) are for estimates of drifting snow transport based on a scaling of the curvature: drifting snow transport (kg m$^{-2}$ yr$^{-1}$) = $\alpha$ ($10^6$ kg m$^{-1}$ yr$^{-1}$) × curvature ($10^{-6}$ m$^{-1}$), with $\alpha = 0$ ($10^6$ kg m$^{-1}$ yr$^{-1}$) for wind speed lower than $ws_{min}$ (m s$^{-1}$), $\alpha = \alpha_{max}$ ($10^6$ kg m$^{-1}$ yr$^{-1}$) for wind speed greater than $ws_{max}$ (m s$^{-1}$), and $\alpha$ linearly increasing as a function of wind speed in between. Wind speed is the annual average of 10 m wind speed of MAR forced by ERA-Interim.

| Component | (3700,5,9) | (3700,6,8) | (4700,5,9) | (2700,5,9) | RACMO2 |
|---|---|---|---|---|---|
| TIS w/o Peninsula | | | | | |
| Mass loss (Gt yr$^{-1}$) | 82 | 81 | 95 | 66 | 21 |
| Mass gain (Gt yr$^{-1}$) | 74 | 74 | 88 | 58 | 16 |
| Net (Gt yr$^{-1}$) | 8 | 7 | 7 | 8 | 5 |
| GIS w/o Peninsula | | | | | |
| Mass loss (Gt yr$^{-1}$) | 81 | 80 | 94 | 65 | 19 |
| Mass gain (Gt yr$^{-1}$) | 68 | 69 | 81 | 53 | 14 |
| Net (Gt yr$^{-1}$) | 13 | 11 | 13 | 12 | 5 |

[Figure]

Figure S1: Snow density modelled by MAR (maps) and observations (dots) for (a) the first 20 cm of snow, (b) the first 50 cm of snow and (c) the first meter of snow, and (d) shows scatterplot of modelled versus observed snow density. The snow density database is detailed in Table  S1. Modelled snow density is taken in average for the period 1979-2015. Observed snow density is averaged on MAR grid cells.

[Figure]

(1) At each observation location:
  · observed SMB over time span
  · bi-linear interpolation of modelled SMB
    (mean on same time span as observation)

(2) For each grid cell containing observation:
  · average of observed and interpolated
    modelled values weighted by the
    observations time span

(3) For transects:
  · choice of a direction (x or y of the
    stereographic grid)
  · average of grid cells orthogonally to this
    direction, weighted by the sum of
    observations time span

Figure S2: Sketch explaining the comparison method between observed (points) and modelled (gridded) SMB.

[Figure]

Figure S3: Estimate of the SMB spatial variability into 35 km×35 km grid cells as a function of mean observed SMB in the grid cell. (a) Standard deviation versus mean value of observed SMB for each MAR grid cell containing more than 10 observations. We delimitate three variability regimes depending on mean SMB values : $<=50$ kg m$^{-2}$ yr$^{-1}$, [50-250] kg m$^{-2}$ yr$^{-1}$ and $>=250$ kg m$^{-2}$ yr$^{-1}$. (b) Location of the SMB regimes, with same colour code as in panel (a).

[Figure]

Figure S4: Difference between mean annual SMB modelled by MAR forced by (a) JRA-55 and (b) MERRA2 and MAR forced by ERA-Interim, for the period 1979-2015, in kg m$^{-2}$ yr$^{-1}$. (c) and (d) are the same than (a) and (b) but divided by MAR(ERA-Interim) mean SMB (in %).

[Figure]

Figure S5: Difference between MAR and RACMO2 forced by ERA-Interim for the period 1979-2015 for (a-c) SMB and (b-d) snowfall. (a-b) Absolute differences, in kg m$^{-2}$ yr$^{-1}$, and (c-d) relative differences, in %. In (a-b), blue lines delimitate areas where the SMB/snowfall difference is 30 % greater than MAR SMB/snowfall, with solid lines when MAR is greater than RACMO2 and dashed lines when MAR is lower than RACMO2.

[Figure]

Figure S6: Annual mean modelled sublimation fluxes for the period 1979-2015, in kg m$^{-2}$ yr$^{-1}$. (a) Sublimation at the surface of the snowpack modelled by MAR(ERA-Interim). (b) Total sublimation (surface snow sublimation plus drifting snow sublimation) modelled by RACMO2(ERA-Interim). (c) Same as (a) but for RACMO2(ERA-Interim). (d) Drifting snow sublimation modelled by RACMO2(ERA-Interim). MAR does not include drifting snow in these simulations.

[Figure]

Figure S7: Snowmelt amounts modelled by MAR and RACMO2 forced by ERA-Interim for the period 1979-2015, in kg m$^{-2}$ yr$^{-1}$. Note that snowmelt is almost totally refrozen in the snowpack in both models (Table 2).

[Figure]

Figure S8: Annual SMB components summed over the Antarctic ice-sheet excluding peninsula (13.4 10$^6$ km$^2$), for (a) SMB, (b) snowfall, (c) sublimation and (d) snowmelt. Red solid thick line is for RACMO2(ERA-Interim), light green solid thin line is for MAR(ERA-Interim), blue solid thick line is for MAR(JRA-55) and dark green solid thin line is for MAR(MERRA2). Note that snowmelt is almost totally refrozen in the snowpack in both models (Table 2).

[Figure]

Figure S9: (a) Curvature of topography computed on the MAR grid ($10^{-6}$ m$^{-1}$) (b) Divergence of the mean annual 10 m wind in MAR (m s$^{-1}$ km$^{-1}$)

[Figure]

Figure S10: (top) Correlation coefficient R between MAR(ERA-Interim) SMB bias and curvature spatially shifted of -2, -1, 0, 1 and 2 grid cells. Green bars are for p-value lower than 0.05 and R greater than 0. (bottom) Scatterplots of MAR(ERA-Interim) SMB bias versus shifted curvature, with shift given at top left of each sub-figure. Pink dashed line is the regression line through origin computed for the four transects all-together (Fig. 4a).  Squares are for locations where MAR annual 10 m wind speed in lower than  7 m s$^{-1}$. For the transect Zhongshan–Dome A, we excluded one data point with low wind speed (square with black outline) and two data points which were clear outliers (dots with black outlines). For the transect Syowa–Dome F, we excluded 5 data points with low wind speed (squares with black outlines).

[Figure]

Figure S11: Estimate of the Coriolis deflection of the katabatic wind flow at the ice sheet surface. We compute the angle between the gradient of the topography (direction of the maximum slope) and the wind direction, and convert it in a deflection value, in percentage of the grid box size (deflection = tan(angle)). As transects are shown from the coast to the plateau, the  deflection  is  counted along this same axis: a  deflection toward the coast shifts the wind  backward in the axis (negative deflection), and a  deflection toward the plateau shifts the wind  upward in the axis (positive deflection). Finally, as curvature of the topography is used as a proxy of wind divergence which drives the drifting snow transport, the shift of  curvature of +/- one grid cell according to the maximum of correlation with SMB bias (Fig. S10) is in agreement with the Coriolis wind deflection.

[Figure]

Figure S12: (a) Atmospheric boundary layer (ABL) moisture holding capacity in MAR for the year 2015, in kg m$^{-2}$ yr$^{-1}$. The ABL moisture holding capacity is computed with daily variables: ABL moisture holding capacity $= \sum_{k=surface}^{k=ABLsummit}(Qsat-Q)\Delta P/g$, with $Q$ the specific humidity, $Qsat$ the specific humidity at saturation, $\Delta P$ the pressure width of the atmospheric layer $k$ and $g$ the gravitational acceleration. We compute the top of the ABL as the level where the turbulent kinetic energy amounts to 1% of the turbulent kinetic energy maximum in the lowest layers of the model (5% is used in Gallée et al., 2015). We compute $Qsat$ using the relative humidity $rh$: $Qsat = Q/rh$. (b) Difference between the ABL moisture holding capacity in MAR and the drifting snow sublimation in RACMO2, for the year 2015, in kg m$^{-2}$ yr$^{-1}$ (c) ABL moisture holding capacity in MAR (blue dots) and drifting snow sublimation in RACMO2 (red dots), for the year 2015, in kg m$^{-2}$ yr$^{-1}$, as a function of the mean 2 m air temperature in MAR, for the year 2015, in °C. The thin solid blue lines are normalised log-normal distribution of the ABL moisture holding capacity in MAR for 5°C temperature bins around -40°C, -30°C, and -20°C. The thick blue dashed line shows the 95% end of the distributions, and the thick blue solid line is the 99% end of the distributions. The pink line shows a Clausius-Clapeyron-like relationship with temperature: $y = exp(-L_s/R_v(1/ta - 1/ta_0) + log(subl_0))$, in kg m$^{-2}$ yr$^{-1}$, with $ta$ the air temperature in K, $L_s$ the enthalpy of sublimation (2.8 10$^6$ J kg$^{-1}$), $R_v$ the gas constant of water vapor (461.52 J kg$^{-1}$ K$^{-1}$), $ta_0 = 263.15$ K and $subl_0 = 500$ kg m$^{-2}$ yr$^{-1}$. (d) Same as (c) but for surface elevation instead of air temperature. Normalised distributions are computed for 500 m bins around 1000 m asl, 2000 m asl, and 3000 m asl. The ABL moisture holding capacity computed in the MAR model represents the maximum moisture amount that can be loaded in the atmospheric boundary layer according to the MAR simulations. We can confidently consider this ABL moisture holding capacity as an upper bound for drifting snow sublimation amounts (panels a and b), as MAR not including the drifting snow process implies that the ABL keeps its full potential to hold moisture. The ABL moisture holding capacity is exponentially dependent to the air temperature, following a Clausius-Clapeyron-like relationship (panel c).

[Figure]

Figure S13: For each of the four long transects is shown, from top to bottom, for the year 2015: (top row) 2 m air temperature, in °C; (2nd row) atmospheric boundary layer moisture holding capacity in MAR (blue line), and drifting snow sublimation in RACMO2 (red line), in kg m$^{-2}$ yr$^{-1}$; (3rd row) drifting snow transport estimate as a function of curvature (black line), and drifting snow transport simulated by RACMO2 (solid red line), in kg m$^{-2}$ yr$^{-1}$; (bottom row) the difference between modelled and observed SMB for MAR (blue line) and RACMO2 (red line), in kg m$^{-2}$ yr$^{-1}$. The blue bands are when the curvature of the topography is greater than 0.004 10$^{-6}$ m$^{-1}$ (crests) and yellow bands are when the curvature of the topography is lower than -0.004 10$^{-6}$ m$^{-1}$ (valleys).